# The Mesozoic terminated in boreal spring

Melanie A. D. During[1,2 ✉], Jan Smit[1], Dennis F. A. E. Voeten[2,3], Camille Berruyer[3], Paul Tafforeau[3], Sophie Sanchez[2,3], Koen H. W. Stein[4,5], Suzan J. A. Verdegaal-Warmerdam[1] & Jeroen H. J. L. van der Lubbe[1,6]

The Cretaceous–Palaeogene mass extinction around 66 million years ago was triggered by the Chicxulub asteroid impact on the present-day Yucatán Peninsula[1,2]. This event caused the highly selective extinction that eliminated about 76% of species[3,4], including all non-avian dinosaurs, pterosaurs, ammonites, rudists and most marine reptiles. The timing of the impact and its aftermath have been studied mainly on millennial timescales, leaving the season of the impact unconstrained. Here, by studying fishes that died on the day the Mesozoic era ended, we demonstrate that the impact that caused the Cretaceous–Palaeogene mass extinction took place during boreal spring. Osteohistology together with stable isotope records of exceptionally preserved perichondral and dermal bones in acipenseriform fishes from the Tanis impact-induced seiche deposits[5] reveal annual cyclicity across the final years of the Cretaceous period. Annual life cycles, including seasonal timing and duration of reproduction, feeding, hibernation and aestivation, vary strongly across latest Cretaceous biotic clades. We postulate that the timing of the Chicxulub impact in boreal spring and austral autumn was a major influence on selective biotic survival across the Cretaceous–Palaeogene boundary.

The Cretaceous-Palaeogene (K–Pg) mass extinction event affected biodiversity with high but poorly understood taxonomic selectivity. Among archosaurs, for example, all pterosaurs and non-avian dinosaurs succumbed in the K–Pg mass extinction, while crocodilians and birds survived into the Palaeogene period[3,4]. Direct consequences of the impact, including impact glass fallout, large-scale forest fires and tsunamis, are geologically documented more than 3,500 km from the Chicxulub impact crater[5–8]. Although direct effects of the impact devastated a vast geographical area, the global mass extinction probably unfolded during its aftermath, which involved rapid climatic deterioration estimated to have lasted up to several thousands of years[9–11]. Whether seasonal timing of the onset of these marked changes affected the selectivity of the K–Pg extinction could not yet be established owing to the lack of suitable records.

The Tanis event deposit in North Dakota (USA) is an exceptional seiche deposit preserving a rich thanatocoenosis (that is, a mass death assemblage) of latest Cretaceous biota at the top of the Hell Creek Formation. The majority of macrofossils encountered at the Tanis locality represent direct casualties of the K–Pg bolide impact that were buried within the impact-induced seiche deposit[5]. Tens of minutes after the impact, the seiche agitated large volumes of water and soil in the estuary of the Tanis river[5]. As the seiche proceeded upstream, it advected bones, teeth, bivalves, ammonites, benthic foraminifera (Extended Data Fig. 1a–c) and plant matter in the suspended load while impact spherules rained down from the sky[5]. Within the thanatocoenotic accumulation, abundant acipenseriforms—sturgeons and paddlefishes—became oriented along the seiche flow directions and buried alive with numerous impact spherules in their gills[5] (Fig. 1, Extended Data Fig. 2a, b).

During the Maastrichtian (that is, the last age of the Cretaceous), the climate of present-day North Dakota involved four seasons that were documented in tree-ring records recovered from other Upper Cretaceous sites in the Hell Creek Formation[12,13]. Tanis was located at approximately 50° N during the latest Cretaceous and experienced distinct seasonality in rainfall and temperature[14]. Regional air temperatures were reconstructed to range from 4–6 °C in winter up to an average of about 19 °C in summer[13,14]. To uncover the season of the K–Pg bolide impact, we analysed osteohistological records of acipenseriform bone apposition in three paddlefish dentaries and three sturgeon pectoral fin spines that were excavated at the Tanis site in 2017 (Extended Data Fig. 1d–j). These skeletal elements preserve unaltered growth records from embryonic development up to death, making them highly suitable for life history reconstructions[15,16].

## Growth records of end-Cretaceous fishes

To trace appositional growth and pinpoint the season in which bone apposition terminated, we first assessed the preservation of bone growth patterns across the studied specimens. We prepared dermal bone slices of six acipenseriform specimens as microscopic slides and subjected these to osteohistological assessment, during which lines of arrested growth (LAGs) were easily recognized (Fig. 2). To corroborate the annual nature of the LAGs using virtual high-resolution osteohistology[17,18], three-dimensional (3D) volumes were produced with propagation phase-contrast synchrotron radiation micro-computed tomography[19] on beamline BM05 of the European Synchrotron Radiation Facility, France. The 3D nature of the synchrotron data enables

[1]Department of Earth Sciences, Faculty of Science, Vrije Universiteit Amsterdam, Amsterdam, the Netherlands. [2]Subdepartment of Evolution and Development, Department of Organismal Biology, Evolutionary Biology Centre, Uppsala University, Uppsala, Sweden. [3]European Synchrotron Radiation Facility, Grenoble, France. [4]Royal Belgian Institute of Natural Sciences, Directorate 'Earth and History of Life', Brussels, Belgium. [5]Earth System Science—AMGC, Vrije Universiteit Brussel, Brussels, Belgium. [6]School of Earth and Environmental Sciences, Cardiff University, Cardiff, UK. ✉e-mail: melanie.during@ebc.uu.se

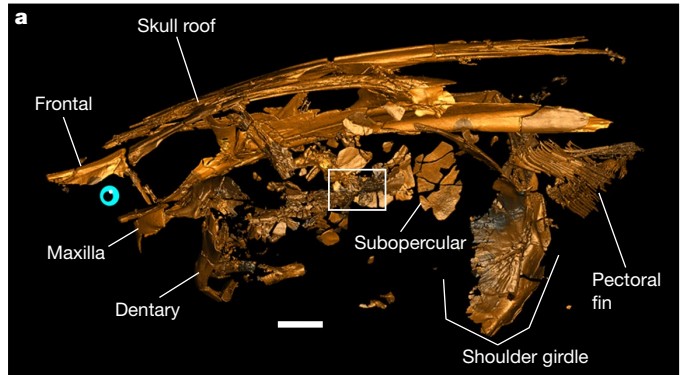

a — Skull roof, Frontal, Maxilla, Dentary, Subopercular, Shoulder girdle, Pectoral fin

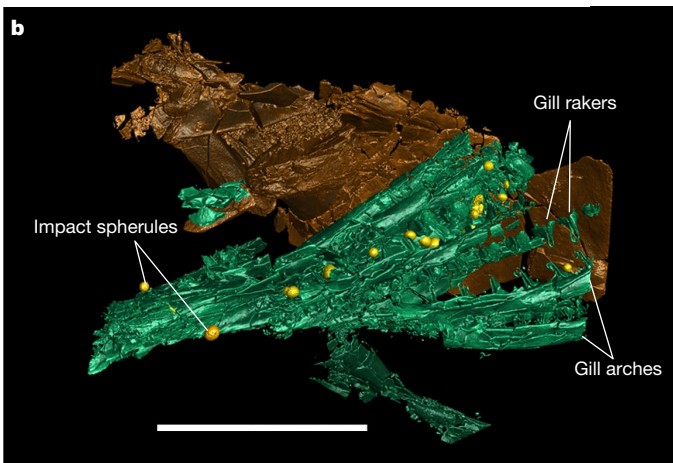

b — Gill rakers, Impact spherules, Gill arches

**Fig. 1 | Reconstruction of a paddlefish with impact spherules in the gill rakers. a**, Three-dimensional rendering of paddlefish FAU.DGS.ND.161.4559.T in left lateral view with the location of a higher-resolution scan (depicted in **b**) indicated (white outline). **b**, Three-dimensional rendering of the subopercular and gills in **a** with trapped impact spherules (yellow). Scale bars, 2 cm. Two-dimensional tomographic data and fully annotated three-dimensional renderings are provided in Extended Data Fig. 2. A three-dimensional animated rendering of FAU.DGS.ND.161.4559.T is provided as Supplementary Video 1.

optimal projection of the bone deposition pattern across multiple cross-sectional planes and resolved the exact relationship between seasonality and cyclical bone apposition in superb detail[20]. In addition, virtual osteohistology allowed us to visualize the seasonal fluctuations of osteocyte lacunar density and volume, which are poorly expressed in the physical 2D thin sections[18] (Fig. 3c, d) . The osteohistological data (Figs. 2, 3, Extended Data Figs. 3–6) were complemented with an incremental carbon isotope record extracted from one of the paddlefish dentaries (VUA.GG.2017.X-2724).

The tomographic data show that impact spherules associated with the paddlefish skeleton are present exclusively in its gill rakers[5] and are absent elsewhere in the preserved specimen (Fig. 1). The absence of impact spherules outside the gill rakers demonstrates that spherules were filtered out of the surrounding waters but had not yet proceeded into the oral cavity or further down the digestive tract, and had not impacted the fish carcases during perimortem exposure. Impact spherule accumulation in the gill rakers and the arrival of the seiche waves must therefore have occurred simultaneously[5], which implies that the acipenseriforms were alive and foraging during the bolide impact and the last minutes of the Cretaceous.

## Well-conserved bone growth archives

The degree of preservation of the sampled acipenseriform bones was assessed using micro-X-ray fluorescence (Methods, Extended

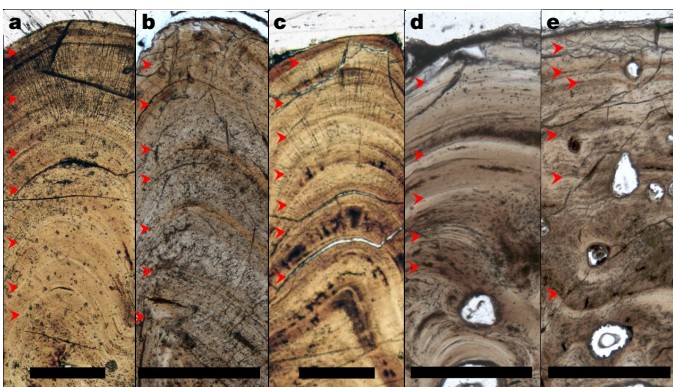

**Fig. 2 | Osteohistological thin sections of five acipenseriform fishes. a–e**, Thin sections in transmitted light of VUA.GG.2017.MDX-3 (**a**), VUA.GG.2017.X-2743M (**b**), VUA.GG.2017.X-2744M (**c**), VUA.GG.2017.X-2733A (**d**) and VUA.GG.2017.X-2733B (**e**), showing congruent pacing of bone apposition during the final years of life, terminating in spring. Red arrows indicate LAGs. Scale bars, 0.5 mm.

Data Figs. 7–9), which would reveal potential taphonomic elemental exchange that may have affected the primary stable isotope composition. The micro-X-ray fluorescence maps show that Fe and Mn oxides are present in the bone vascular canals and surrounding sediments (Extended Data Fig. 8), but have not invaded the bone apatite ($Ca_5(PO_4, CO_3)_3(OH,F,Cl)$). Detrital components, characterized by high concentrations of K and Si, remain restricted to the sediment matrix (Extended Data Fig. 8f–j). The bone apatite conserves a highly homogeneous distribution of P and Ca (Extended Data Fig. 9), which corroborates the unaltered preservation of these apatitic tissues. Skeletal remains of the paddlefishes and sturgeons thus experienced negligible diagenetic alteration, probably as a consequence of rapid burial and possibly aided by early Mn and Fe oxide seam formation[21,22]. The exquisite 3D preservation of delicate structures, including non-ossified tissues that originally enveloped the brain (Extended Data Fig. 2c–f), further demonstrates the excellent preservation of the fossils and absence of taphonomic reorganization[23].

## Consistent records of a spring death

Paddlefish dentaries form through perichondral ossification around the Meckel's cartilage[24]. Sturgeon pectoral fin spines consist of dermal bone—an intramembranous skeletal tissue that forms in the mesenchyme (mesodermal embryonic tissue)[25]. Unlike endochondral bone, perichondral and dermal bone do not originate through mineralization of cartilaginous precursors[26–28] but grow exclusively through incremental bone matrix apposition by secretion of a row of osteoblasts[24,26–28]. The thickness of one annual growth mark cumulatively spans a thick (favourable) growth zone, a thinner (slowly deposited) annulus and, ultimately, a LAG[20]. Our microscopic and virtual osteohistological data consistently show that the six fishes perished (that is, stopped growing) while forming a growth zone shortly after a LAG was deposited (Figs. 2, 3, Extended Data Figs. 3–6), which coincides with an early stage of the favourable growth season[20]. The outermost cortices of all six acipenseriform individuals studied here also exhibit increasing osteocyte lacunar densities and sizes towards their periosteal surfaces (Fig. 3c, Extended Data Figs. 5, 6). In all specimens, this density remained lower than the highest densities and average sizes recorded in previous years (Fig. 3c, Extended Data Figs. 3–6, 10b). As osteocyte lacunar density and size patterns were consistently cyclical across the preceding years during which they peaked at the climaxes of the growth seasons, the last recorded growth season had thus not yet climaxed at the time of death (Figs. 2, 3, Extended Data Figs. 3–6, 10b).

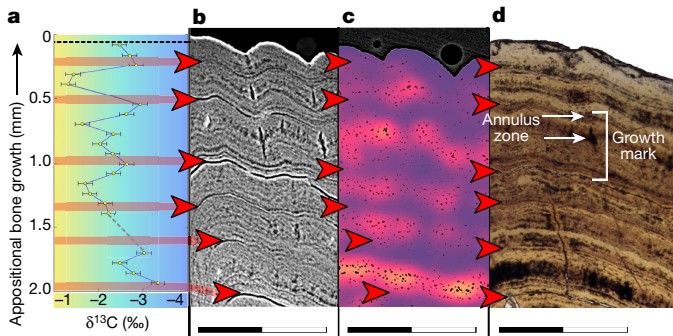

**Fig. 3 | Carbon isotope record alongside the incremental growth profiles across the dentary of paddlefish VUA.GG.2017.X-2724. a**, $\delta$ expressed as ‰ on the Vienna Pee Dee Belemnite (VPDB) reference scale. The colour gradient highlights the theoretical range between maximum values during seasonal (summer) trophic increase of $^{13}$C (yellow) and minimum values during trophic decrease of $^{13}$C (winter) (blue). **b**, Virtual thick section (average-value projection with 0.1-mm depth) showing growth zones during the favourable growth seasons and annuli and LAGs outside the favourable growth seasons. **c**, Cell density map[51] of a virtual thick section (minimum-value projection with 0.2-mm depth) showing fluctuating osteocyte lacunar densities and sizes, with higher densities and largest sizes recorded during the favourable growth seasons (orange) and lower densities and smaller sizes outside the favourable growth seasons (purple). A comparative image of a larger section of bone with scale is provided in Extended Data Fig. 6. **d**, Microscopic thin section in transmitted light showing LAGs (red arrows) and a single growth mark indicated (bracket) spanning the distance between two subsequent LAGs and including a zone and an annulus (Extended Data Fig. 10b). Scanning data visualized in **b** and **c** were obtained approximately 10-mm distal from the physically sectioned thin slice of **d**, which itself was located directly proximal to the thick section sampled for **a**. Scale bars, 1 mm. Corresponding osteohistological data of the other five sampled acipenseriform fishes are presented in Extended Data Figs. 3–5.

The inferred annual growth cycles are independently corroborated by a stable carbon isotope ($\delta^{13}C_{sc}$) archive that recorded several years of seasonal dietary fluctuations in growing bone. Paddlefish VUA. GG.2017.X-2724 also yielded, in addition to this $\delta^{13}C_{sc}$ archive, an oxygen isotope ($\delta^{18}O_{sc}$) record across the final six years of its life (Supplementary Data Table 1, Extended Data Fig. 10a, Methods). The low and constant $\delta^{18}O_{sc}$ values in VUA.GG.2017.X-2724 reflect exclusive inhabitation of freshwater environments by the paddlefishes. This implies that their osteohistological records must have captured seasonal variability rather than, for example, migration between saline and freshwater habitats. Although modern sturgeons are known to have anadromous lifestyles[29,30], this remains to be confirmed for the fossil sturgeons at Tanis, as isotopic data from sturgeon pectoral fin spines could not be secured (Methods, 'Micromill'). Notably, the osteohistological records of all our sturgeons and paddlefishes converge on the same annual growth phase, despite their potential different lifestyles.

Like their modern-day relatives, the latest Maastrichtian paddlefishes of Tanis were filter feeders that presumably consumed copepods and other zooplankton[29–31]. These fishes probably experienced an annual feeding pattern, determined by fluctuating food availability, that peaked between spring and autumn[31]. During maximum productivity, ingested zooplankton enriches the growing skeleton of filter-feeding fishes with $^{13}$C relative to $^{12}$C[32,33]. Thus, the cyclically elevated $^{13}$C/$^{12}$C ratios in paddlefish VUA.GG.2017.X-2724 (Fig. 3a) reflect distinct episodes of high food availability and consumption. Carbon isotope records across the growth record of Paddlefish VUA.GG.2017.X-2724 indicate that peak annual growth rate was not yet attained and the feeding season had thus not yet climaxed—corroborating a boreal spring death.

## Implications for selective K–Pg survival

The Chicxulub bolide impact caused a global heat pulse that ignited widespread wildfires[9,34]. After this heat wave, the last boreal spring of the Mesozoic transitioned to a global impact winter[10]. Although a June timing for the K–Pg impact has been suggested on the basis of palaeobotanical indications for anomalous freezing in this region (Wyoming, USA)[35], the palaeobotanical identities, taphonomic inferences and stratigraphic assumptions underlying that conclusion have since all been refuted[36–39]. Moreover, post-impact cooling happened in the first months to decades following the K–Pg impact[10], which renders proxies registering post-impact freezing conditions asynchronous with the impact event itself.

A suite of impact-induced phenomena contributed to the K–Pg extinction on differing timescales[40,41]. In the days to months following the impact, its instantaneous effects, such as intense infrared radiation caused by ejecta reentry[34], resulting wildfires[9,34] and the spread of sulfurous aerosols leading to acid precipitation[42] must have predominantly afflicted the exposed continental environments. Although negotiating these hostile conditions would not have guaranteed survival, an early clade-wide eradication would always have meant immediate extinction[41].

The seasonal timing of the catastrophic end-Cretaceous bolide impact places the event at a particularly sensitive stage for biological life cycles in the Northern Hemisphere. In many taxa, annual reproduction and growth take place during spring. Species with longer incubation times, such as non-avian reptiles, including pterosaurs and most dinosaurs, were arguably more vulnerable to sudden environmental perturbations than other groups[43] (for example, birds). Southern Hemisphere ecosystems, which were struck during austral autumn, appear to have recovered up to twice as fast as Northern Hemisphere communities[44], consistent with a seasonal effect on biotic recovery.

Subterranean sheltering conceivably contributed to the cynodont survival of the Permo-Triassic (PT) crisis[45]. Similarly, large-scale wildfires raging across the Southern Hemisphere[9,34,41] may have been evaded by hibernating mammals that were already sheltered in burrows[34,41] in anticipation of austral winter. Additional modes of seasonal dormancy, torpor and/or aestivation, which are nowadays practised by various mammals[46,47] as well as certain amphibians, birds and crocodilians[48], could have facilitated further underground survival. In the aftermath of the K–Pg event, ecological networks collapsed from the bottom up. Floral necrosis[9] and extinction immediately affected species dependent on primary producers, while some animals capable of exploiting alternative resources—for example, certain birds and mammals[49,50]—persisted.

## Conclusions

Seasonal timing of the Chicxulub impact in boreal spring and austral autumn will aid in further calibrating evolutionary models exploring the selectivity of the K–Pg extinction and the asymmetry in extinction and recovery patterns between the two hemispheres. Decoupling short- and long-term effects of the bolide impact on the K–Pg mass extinction will also aid in identifying extinction risks and modes of ecological deterioration caused by the forthcoming global climate change. The uniquely constrained Tanis site[5] offers valuable proxies for reconstructing the environmental, climatological and biological conditions that prevailed locally when the Mesozoic ended.

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

# Methods

## Fieldwork

Excavation at the Tanis locality in south-western North Dakota took place between 10 August and 20 August 2017. Sections of dentaries of paddlefishes and pectoral fin spines of sturgeons were collected in the field for histological study.

## Thin sectioning

Four out of the six samples were excavated from the sediment matrix. These included all sturgeon pectoral fin spines (VUA.GG.2017.X-2743M, VUA.GG.2017.X-2744M, and VUA.GG.2017.MDX-3) and one of the paddlefish dentaries (VUA.GG.2017.X-2724). Paddlefish dentaries VUA. GG.2017.X-2733A and VUA.GG.2017.X-2733B, belong to two individuals, were uncovered aligned to each other and fractured upon discovery. To avoid further damage, the samples were embedded in epoxy resin prior to thin sectioning. All specimens were cut with a diamond saw and polished to obtain microscopic thin sections (about 50-µm thick) and thick sections for micromilling (about 200-µm thick). See Extended Data Fig. 1e–j for images of the specimens and the sampling locations.

## Osteohistological analysis

In the acipenseriform dermal bones examined in this study, annual growth cyclicity can be traced through growth marks (GMs).

A GM spans a single growth cycle that typically lasts one year and can be divided into a zone, an annulus, and a LAG[20,52]. The zone is deposited during a period of relative rapid growth in the active or favourable growth season[20]. The annulus is subsequently formed when growth slows down towards the end of the growth season[20]. Finally, a LAG forms when growth periodically ceases until the next growth season starts and a new zone is deposited[20].

During the formation of a growth zone, the density and volumes of osteocyte lacunae (OL; subcircular dark features in Extended Data Fig. 10a) initially increase when growth accelerates. Subsequently, towards and into the annulus, OL density and volume decrease as growth slows down[18]. Because a LAG coincides with a temporary arrest of local osteogenesis, it is only expressed when deposition of a new growth zone has commenced. All six studied specimens show a LAG relatively close to the outermost partial growth zone.

In fossil bone, LAGs often appear as sharply defined dark lines[53] that typically constitute a poorly coherent interface between adjacent bone layers, thus facilitating (local) delamination between adjacent cortical layers[53]. During fossilization, percolation products can accumulate in these gaps and thereby (locally) accentuate the LAGs[51,53] (figure 31.3G of ref. [52]). Based on this well-understood expression of LAGs (that we recognize from our own experience as well; S.S. personal observation), we have consistently identified the LAGs as locally stained dark lines that may be associated with circumferentially propagated cracked surfaces which are oriented parallel to the periosteal deposits.

Besides cyclical seasonal factors that synchronize GM accretion, stress may induce additional diapause stages that result in supplementary marks within a single year[54]. Cessation of growth for the duration of several weeks can provoke the formation of a LAG[54]. However, such non-cyclical marks "tend to be haphazard rather than regular (that is, they do not reflect a particular spacing or rhythm)" and do not encircle the cortex of the skeletal element but "tend to be locally confined to an arc"[55].

As the studied bones yield only regularly spaced GMs along their complete circumference, we confidently identify the preserved GMs as annual cycles. Moreover, the fluctuating quantified density and volumes of osteocyte lacunae (Extended Data Fig. 6d–f) and the carbon isotopic record (Fig. 3a, Extended Data Fig. 10a) across the final seven years of growth of VUA.GG.2017.X-2724 are exclusively consistent with the identification of annual LAGs in corresponding physical thin sections. In all studied specimens, bone growth terminated during the process of zonal bone growth.

## Micro-X-ray fluorescence

Fragments of the paddlefish and sturgeon samples that remained after thin sectioning were analysed with microX-ray fluorescence. High-resolution elemental mapping was conducted using a Bruker M4 Tornado 2D spectrometer at 50 kV and 600 µA, without a filter, and at an acquisition rate of 20 µm per 5 ms at the Vrije Universiteit Brussel.

## Micromill

The growth increments were sampled in the thick sections (about 200-µm thick) at the highest possible accuracy using a Micromill (Merkantek). Drill transects were assigned in the accompanying software and after each individual sample was collected, the drill bit was cleaned with ethanol. Not all thick sections were suitable for micromilling. The lobed anatomy of the sturgeon fin spines (VUA.GG.2017.X-2743M and VUA.GG.2017.X-2744M) proved too complex to reliably sample single growth increments with the micromill. Paddlefish dentaries VUA. GG.2017.X-2733A and VUA.GG.2017.X-2733B only exposed a few growth lines that were too narrow to sample with the micromill. Sturgeon pectoral fin spine VUA.GG.2017.MDX-3 and paddlefish dentary VUA. GG.2017.X-2724 were sampled up to the outermost growth increment.

## Stable isotope analysis

Micromilled hydroxyapatite samples of specimen VUA.GG.2017.X-2724 weighing about 50 µg were placed in Exetainer vials (Labco) and flushed with purified helium gas. For reference, the analysed amounts of structural carbonate are equivalent to anout 5 µg of $CaCO_3$. Orthophosphoric acid was subsequently added and allowed to react for 24 h at 45 °C. VUA.GG.2017.MDX-3 was routinely analysed with a Thermo Finnigan Delta[plus] mass spectrometer connected to a Thermo Finnigan GasBench II at the Earth Sciences Stable Isotope Laboratory (Vrije Universiteit, Amsterdam). However, the amount of $CO_2$ generated was found to be too small to permit reliable isotopic determinations. To alleviate this, the GasBench was provisionally interfaced with a cold trap in which the $CO_2$ was frozen with liquid nitrogen during a 2 min period. After trapping for 2 min, an accurate single-pulse measurement was performed for each of the apatitic samples and standards. Each isotopic sample determination was preceded by six pulses of monitoring $CO_2$ with a calibrated isotopic composition to assure stable conditions of the mass spectrometer. The isotopic measurements of the weighted micromilled samples were bracketed by the analyses of the inter-laboratorial apatite standard (Ag-Lox) to account for the linearity effect[56]. After corrections, the uncertainties for $\delta^{13}C$ and $\delta^{18}O$ of the Ag-Lox ($n = 4$) were 0.16 ‰ and 0.39 ‰ (1 s.d.) respectively. Although the amount of extracted and analysed structural carbonate remains insufficient for optimal isotopic determination, the relatively large recovered $\delta^{13}C$ variability still yields a meaningful record across the appositional bone archive. The $\delta^{18}O$ values of structural carbonate, unlike those of phosphate ($PO_4$)[57], do not offer a sensitive palaeo-environmental proxy for accurate seasonal temperature reconstructions[58]. However, the relatively constant $\delta^{18}O$ values of structural carbonate precludes large $\delta^{18}O$ changes in ambient water, such as shifts between freshwater and saline environments.

## Propagation phase-contrast synchrotron radiation micro-computed tomography

Paddlefish specimen FAU.DGS.ND.161.4559.T lacks the paddle-shaped rostrum and all aspects caudal to the pectoral girdle. FAU.DGS. ND.161.4559.T was provided by the Palm Beach Museum of Natural History. Data acquisition took place in May 2018 on Beamline BM05 of the European Synchrotron Radiation Facility, Grenoble, France[59]. The complete specimen was scanned at an average energy of 132 keV using the white beam of BM05 filtered with 0.4 mm of Mo and 9 mm of Cu. The detector was composed of a 2-mm-thick LuAG:Ce scintillator optically coupled to a PCO edge 4.2 CLHS sCMOS camera. The resulting voxel size was 43.5 µm. To obtain sufficient propagation phase

contrast, the distance between the sample and the detector was set at 5 m. A total of 205 scans, each consisting of 5,000 projections taken at 7-ms intervals, were performed with a vertical displacement of 1.4 mm at a vertical field of view of 2.8 mm to ensure a double scan of the complete samples. Scans were performed in half-acquisition mode to enlarge the lateral field of view. The volume was reconstructed using a single-distance phase retrieval algorithm coupled with filtered back projection as implemented in the ESRF software PyHST2. Vertical concatenation, 16-bit conversion, and ring artefact corrections were performed using MATLAB scripts developed in-house. The gill region and impact spherules were subsequently scanned at a voxel size of 13.67 μm (filters: 0.4 mm of Mo and 6 mm of Cu, scintillator: LuAG:Ce, 500-μm thick, detected energy: 166 keV, propagation distance: 2.5 m). The samples were scanned in half-acquisition mode in two columns of 77 scans, each consisting of 4,998 projections with exposure times of 0.05 s, that were laterally concatenated after reconstruction. Finally, sample (VUA.GG.2017.X-2724) from the paddlefish dentaries and (VUA. GG.2017.MDX-3, VUA.GG.2017.X-2743M and VUA.GG.2017.X-2744M) of the sturgeon pectoral fin spines were scanned at 4.35 μm voxel size for osteohistological analysis[60] (filters: 3.5 mm of Al plus 11 bars Al with a diameter of 5 mm, scintillator: LuAG:Ce scintillator, 500-μm thick, detected energy: 92 keV, propagation distance: 1.5 m). The samples were scanned in half-acquisition mode in one single column of 22 scans, each consisting of 4,998 projections with exposure times of 60 ms.

Digital 3D extraction of the bones and impact spherules was performed in VGStudio MAX 3.2 (Volume Graphics). VGStudio MAX 3.2 furthermore enabled the creation of virtual thick sections of the osteohistological samples through the 'thick slab-mode', which captures the maximum, average, or minimum, grey-level values along the desired field depth. Virtual thick sections were obtained from the average grey-level values at a thickness of 100 μm following optimal 3D alignment of the annuli and LAGs. Additional virtual thick sections were created from the minimum grey-level values at a thickness of 200 μm to best resolve the sizes and distributions of osteocyte lacunae. A coloured map of the density of the osteocyte lacunar distribution was created with a Gaussian filter[51]. Finally, we visualized the annual cyclicity of osteocyte lacunar volumes[18] in paddlefish dentary VUA. GG.2017.X-2724. As the resolution of our data (voxel size of 4.35 μm; appropriate for assessing GMs and osteocyte lacunar distributions) is sixfold lower than that used for earlier osteocyte lacunar volumetric quantification in fish bones[18] (voxel size of 0.7 μm), our result should be considered with appropriate care. Closely spaced (large) osteocyte lacunae may occasionally be conjoined and additional phenomena in the broad size range of osteocyte lacunae may be incidentally included in the visualized distribution. Moreover, in tomographic data, osteocyte lacunae are delimited by slight colour gradients (rather than discrete lines) that scale with voxel size. Because the outermost feature fringe contributes disproportionally to recovered volumes, these values are somewhat skewed relative to the original osteocyte lacunar volumes, which likely produces exaggerated volume values. Therefore, although all rendered features were extracted with a single thresholding operation and relative patterns are conservatively retained, absolute volume values are best considered in a comparative context.

## Reporting summary

Further information on research design is available in the Nature Research Reporting Summary linked to this paper.

## Data availability

All isotopic, geochemical, and osteohistological data are included in the paper and Extended Data. Tomographic data of FAU.DGS.ND.161.4559.T, VUA.GG.2017.X-2724, VUA.GG.2017.MDX-3, VUA.GG.2017.X-2743M, and VUA.GG.2017.X-2744M are available at https://doi.org/10.5281/zenodo.5776294 and the http://paleo.esrf.eu database.

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

**Acknowledgements** M.A.D.D. was partially funded by an EAVP Research Grant (ERG) awarded by the European Association of Vertebrate Palaeontologists. D.F.A.E.V. gratefully acknowledges support from the Wenner-Gren Foundation through fellowships UPD2018-250 and UPD2019-0076. VGStudio Max (Volume Graphics, Germany) and the Porosity/Inclusion Analysis module were funded by the Vetenskapsrådet through grants 2015-04335; 2019-04595 to S.S. We thank R. DePalma for providing guidance in the field and access to the specimens. We acknowledge the ESRF for provisioning beamtime at BM05. We thank V. Fernandez and K. Chapelle for their assistance with the segmentation in VGStudio; B. Lacet for help with the preparation of the thin and thick sections; M. Hagen for the use of her sedimentology laboratory and the microbalance for weeks in a row; F. Peeters for assistance in photographing the thin sections while sharing his thoughts on the project; and P. Ahlberg for his advice, labelling of the paddlefish bones, fruitful discussions and invaluable consultation.

**Author contributions** M.A.D.D., J.S. and H.J.L.v.d.L. conceived and designed the project. Materials were excavated by M.A.D.D. in 2017. M.A.D.D., D.F.A.E.V., C.B. and P.T. performed the synchrotron experiments. K.H.W.S. and M.A.D.D. performed the micro X-ray fluorescence analysis. M.A.D.D. sampled the specimens with the micromill. M.A.D.D., S.J.A.V.-W. and H.J.L.v.d.L performed the isotope analyses. P.T. processed and reconstructed the raw propagation phase contrast synchrotron radiation micro computed tomography scanning data. M.A.D.D. and D.F.A.E.V. segmented the scanning data. M.A.D.D., J.S., D.F.A.E.V., S.S. and H.J.L.v.d.L. analysed the data. S.S. created Fig. 3c and Extended Data Fig. 8a–c. D.F.A.E.V. created Extended Data Fig. 8d–f. M.A.D.D. created all other figures. All authors discussed the interpretations. M.A.D.D., D.F.A.E.V. and H.J.L.v.d.L. wrote the manuscript. All authors provided a critical review and approved the final draft of the manuscript.

**Funding** Open access funding provided by Uppsala University.

**Competing interests** The authors declare no competing interests.

**Additional information**
**Correspondence and requests for materials** should be addressed to Melanie A. D. During.

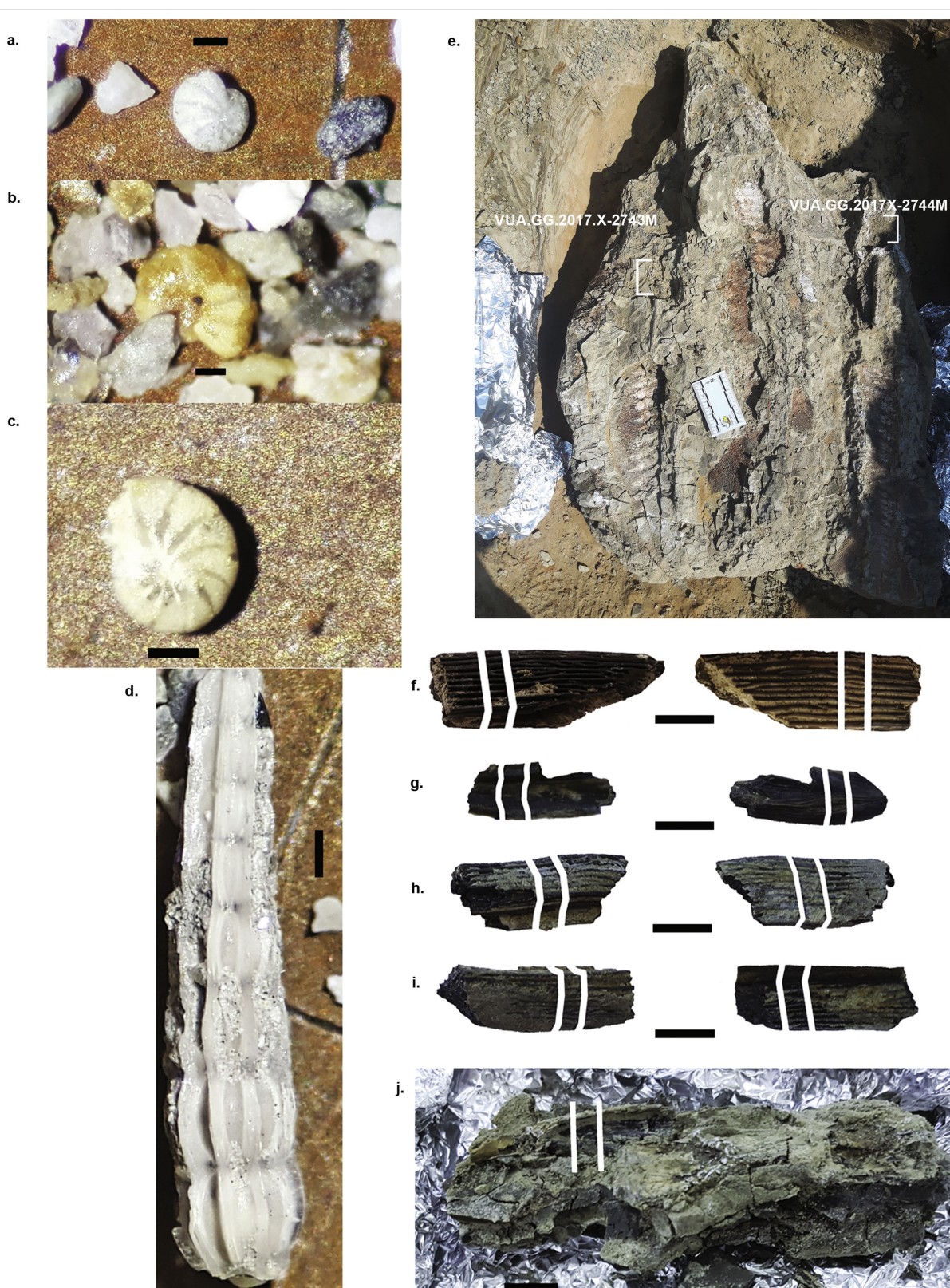

**Extended Data Fig. 1 | Benthic foraminifera and acipenseriforms in the field at the Tanis locality and their respective sampling locations. a,b,c,** and **d,** undetermined benthic foraminifera recovered from the Tanis deposit. Scale bars 1 mm. **e.** Sturgeon pectoral fin spines VUA.GG.2017.X-2743M and VUA. GG.2017.X-2744M, preserved in anatomical position, relative to their respective sturgeon carcasses in the Tanis deposit. **f,** Perichondral sturgeon bone sample VUA.GG.2017.MDX-3. **g,** perichondral sturgeon bone sample VUA. GG.2017.X-2743M. **h,** Perichondral sturgeon bone sample VUA.

GG.2017.X-2744M. **i,** Dermal paddlefish bone sample VUA.GG.2017.X-2724. **j,** VUA.GG.2017.X-2733A in left lateral (left) and right lateral (right) view (sampling locations indicated between lines). The sediment matrix of sample VUA.GG.2017.X-2733A (bottom) also contains specimen VUA. GG.2017.X-2733B (covered). The fragile bone-bearing matrix was stabilised in epoxy resin prior to cutting, which obscured VUA.GG.2017.X-2733B from view. Scale bars 1 cm.

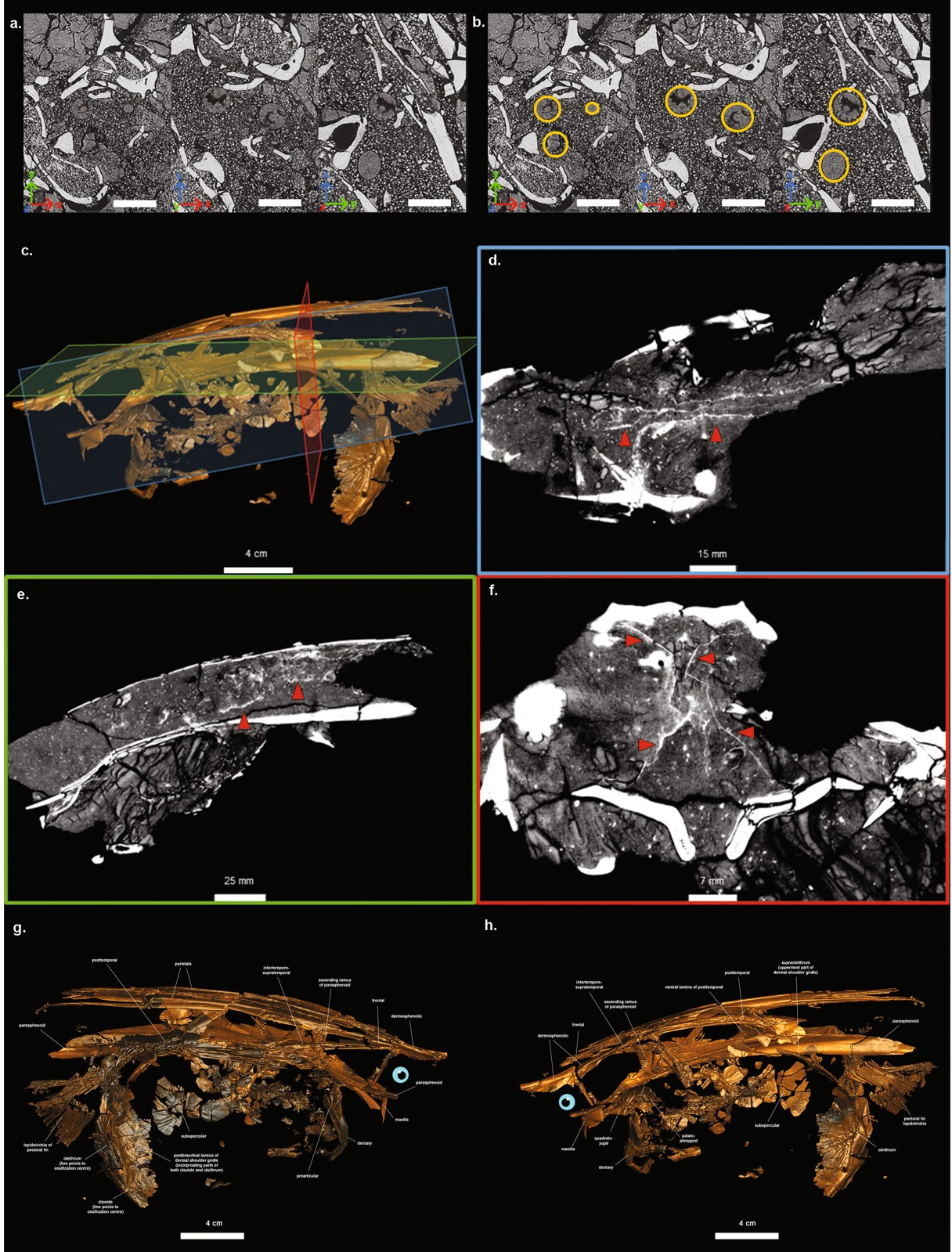

**Extended Data Fig. 2 | PPC-SRµCT data of FAU.DGS.ND.161.4559.T, a partial paddlefish from the Tanis locality. a**, Orthogonal virtual thin sections (100 µm thick, average-value projections) obtained in front, top, and right view. **b**, Impact spherules in virtual thin sections of **a**, indicated with yellow circles. Scale bars **a** 1 mm. **c**, Three-dimensional rendering (in left lateral view) with virtual cross sections of **d** (blue), **e** (green), and **f** (red) indicated. **d**, Coronal virtual slice. **e**, Sagittal virtual slice. **f**, Axial virtual slice, brain-enveloping tissues indicated with red arrows. **g**, Three-dimensional rendering in right lateral view with anatomical labels. **h**, Three-dimensional rendering in left lateral view with anatomical labels.

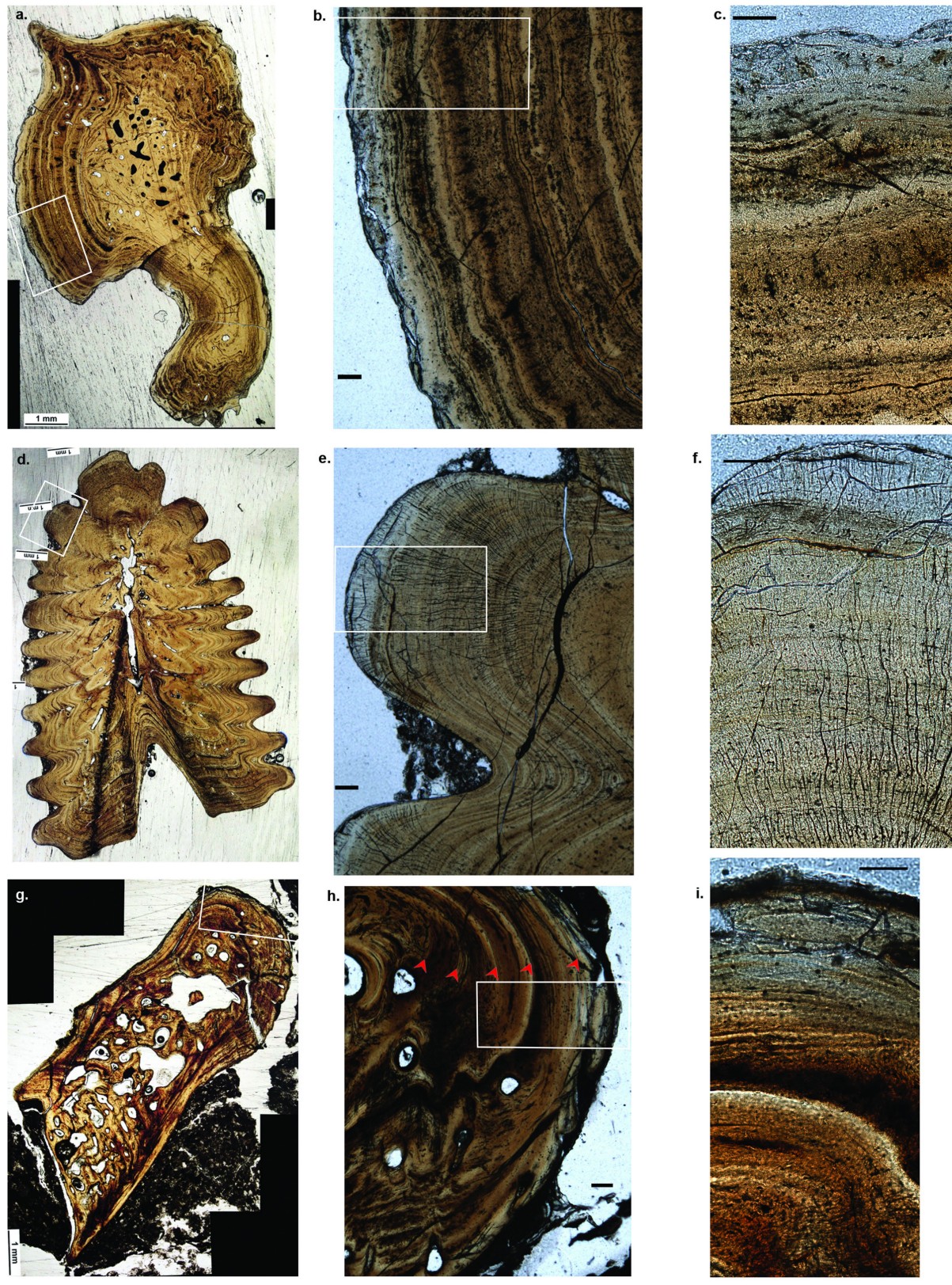

**Extended Data Fig. 3 | Osteohistology of acipenseriforms from the Tanis locality. a**, Thin section of paddlefish dentary VUA.GG.2017.X-2724 under transmitted light. **b**, Detail of VUA.GG.2017.X-2724 thin section (white box in **a**), scale bar 100 µm. **c**, Detail of VUA.GG.2017.X-2724 thin section (white box in **b**), scale bar 100 µm. **d**, Thin section of sturgeon pectoral fin spine VUA.GG.2017. MDX-3 under transmitted light. **e**, Detail of VUA.GG.2017.MDX-3 thin section (white box in **d**), scale bar 100 µm. **f**, Detail of VUA.GG.2017.MDX-3 thin section (white box in **e**), scale bar 100 µm. **g**, Thin section of paddlefish dentary VUA. GG.2017.X-2733A under transmitted light. **h**, Detail of VUA.GG.2017.X-2733A thin section (white box in **g**) with red arrows indicating Lines of Arrested Growth (LAGs), scale bar 100 µm. **i**, Detail of VUA.GG.2017.X-2724 thin section (white box in **h**), scale bar 100 µm.

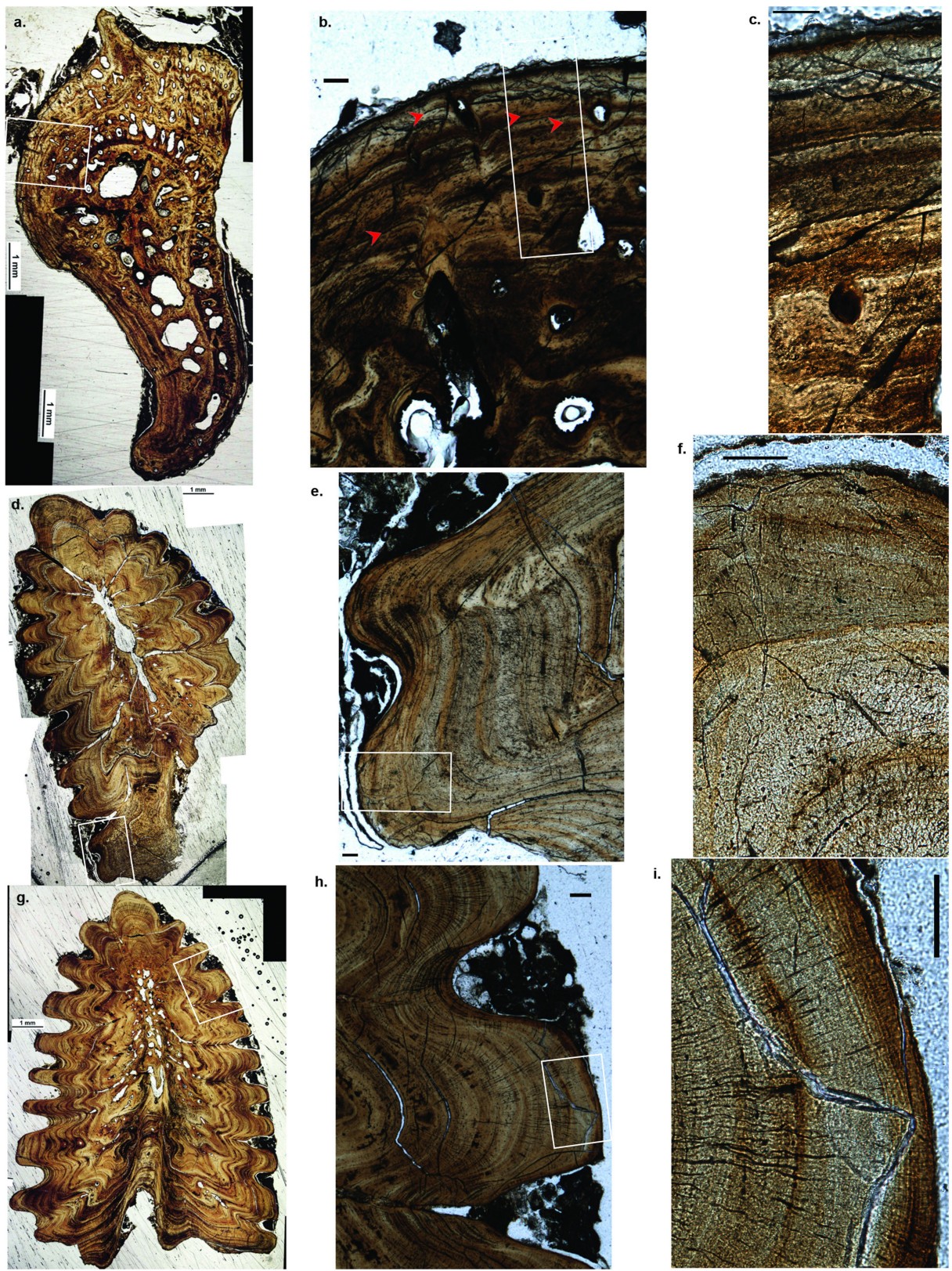

**Extended Data Fig. 4 | Osteohistology of acipenseriforms from the Tanis locality. a**, Thin section of paddlefish dentary VUA.GG.2017.X-2733B under transmitted light. **b**, Detail of VUA.GG.2017.X-2733B thin section (white box in **a**) with red arrows indicating Lines of Arrested Growth (LAGs), scale bar 100 μm. **c**, Detail of VUA.GG.2017.X-2733B thin section (white box in **b**), scale bar 100 μm. **d**, Thin section of sturgeon pectoral fin spine VUA.GG.2017.X-2743M under transmitted light. **e**, Detail of VUA.GG.2017.X-2743M thin section (white box in **d**), scale bar 100 μm. **f**, Detail of VUA.GG.2017.X-2743M thin section (white box in **e**), scale bar 100 under transmitted light. **g**, Thin section of sturgeon pectoral fin spine VUA.GG.2017.X-2744M under transmitted light. **h**, Detail of VUA.GG.2017.X-2744M thin section (white box in **g**), scale bar 100 μm. **i**, Detail of VUA.GG.2017.X-2744M thin section (white box in **h**), scale bar 100 μm.

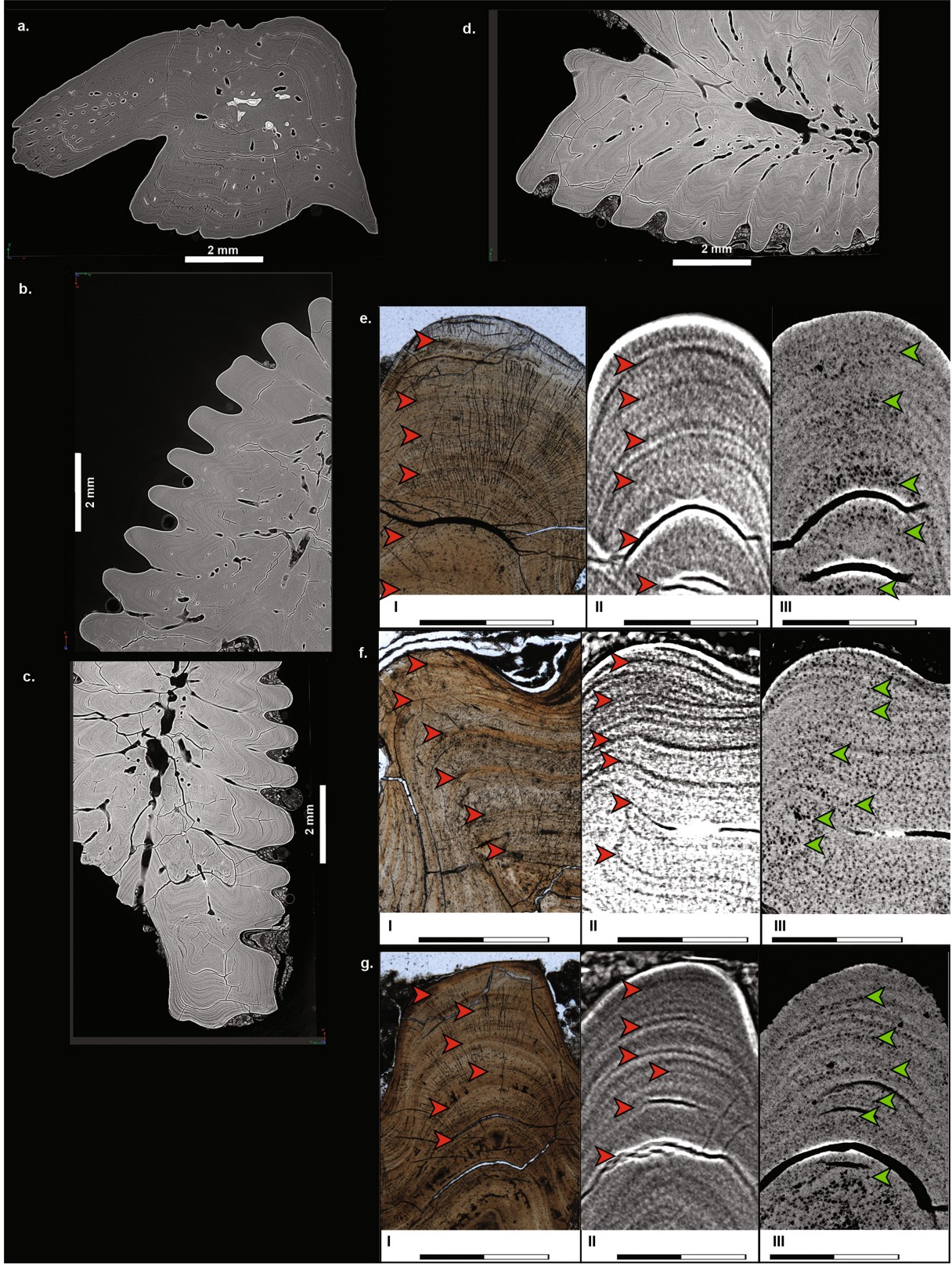

**Extended Data Fig. 5** | See next page for caption.

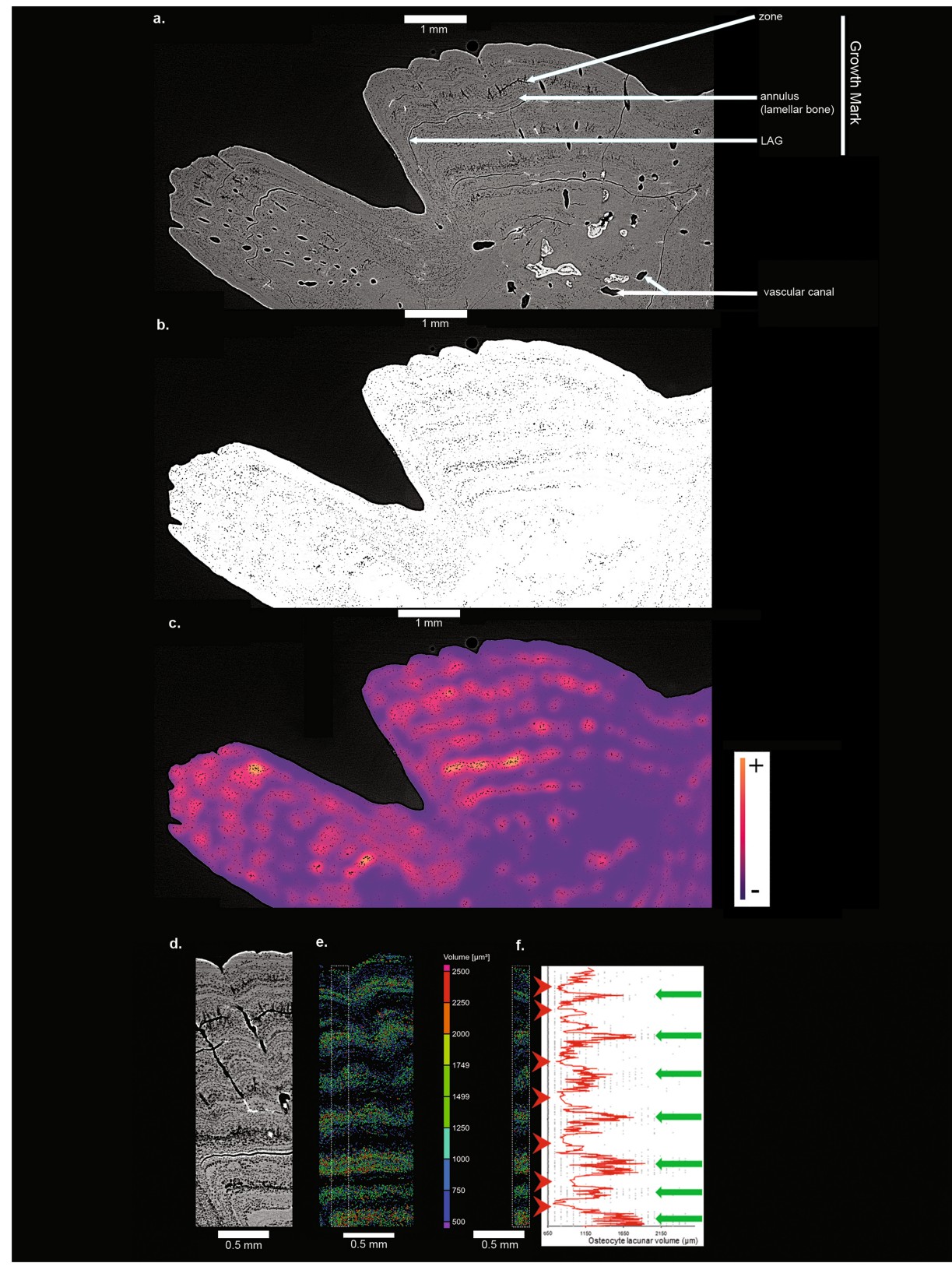

**Extended Data Fig. 6** | See next page for caption.

**Extended Data Fig. 6 | Osteocyte lacunar density and volume distribution in paddlefish dentary VUA.GG.2017.X-2724 revealed by PPC-SRμCT.** **a**, Virtual thin section (100 μm thick, average-value projection) with osteohistological features indicated. **b**, Segmented osteocyte lacunar distribution (black dots). **c**, Osteocyte lacunar density ma 54= Sanchez et al., 2013 p[54] with gradient scale. **d**, Virtual thick section (100 μm thick, minimum-value projection). **e**, Three-dimensional osteocyte lacunar distribution at **d** (depth circa 1230 μm), colour-coded by volume. **f**, Oscillating osteocyte lacunar volumes towards periosteal margin (10-point moving average) in dashed box in **e**. Successive annual growth climaxes (green arrows) and growth cessations (LAGs; red arrows) indicated. Typical annual maximum osteocyte lacunar volumes and highest osteocyte lacunar densities were not yet achieved in the year of death, indicating that growth ceased prior to the annual growth climax projected to occur during summer. Because the data resolution (voxel size of 4.35 μm: appropriate for assessing growth marks and osteocyte lacunar distributions) is sixfold lower than that demonstrated in detailed osteocyte lacunar volume reconstructions[18] (voxel size of 0.7 μm), these results should be considered qualitatively.

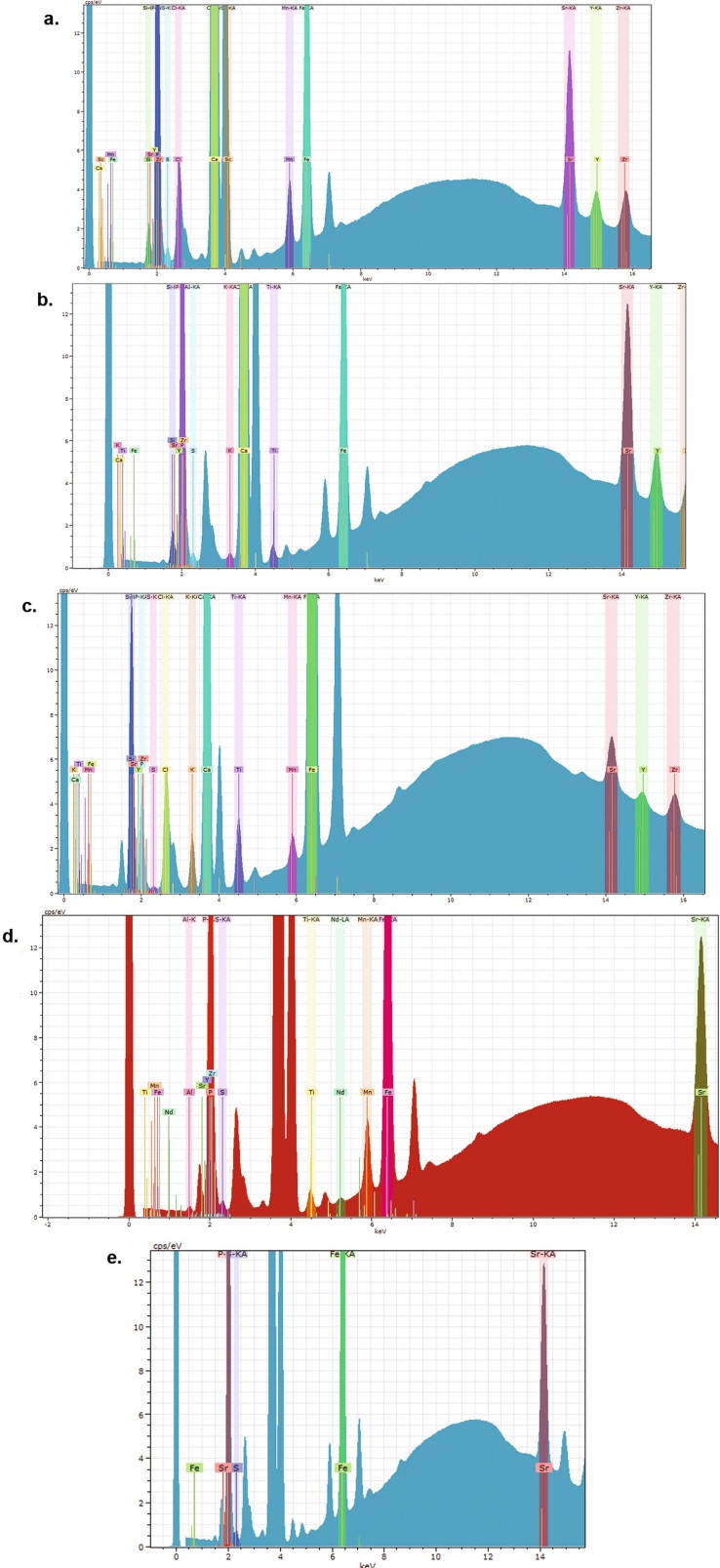

**Extended Data Fig. 7 | Micro-X-ray fluorescence spectra of acipenseriform elements from the Tanis locality. a**, Elemental spectrum of paddlefish dentary VUA.GG.2017.X-2724. **b**, Elemental spectrum of sturgeon pectoral fin spine VUA.GG.2017.MDX-3. **c**, Elemental spectrum of paddlefish dentary VUA. GG.2017.X-2733A, VUA.GG.2017.X-2733B, and surrounding matrix. **d**, Elemental spectrum of sturgeon pectoral fin spine VUA.GG.2017.X-2743M. **e**, Elemental spectrum of sturgeon pectoral fin spine VUA.GG.2017.X-2744M.

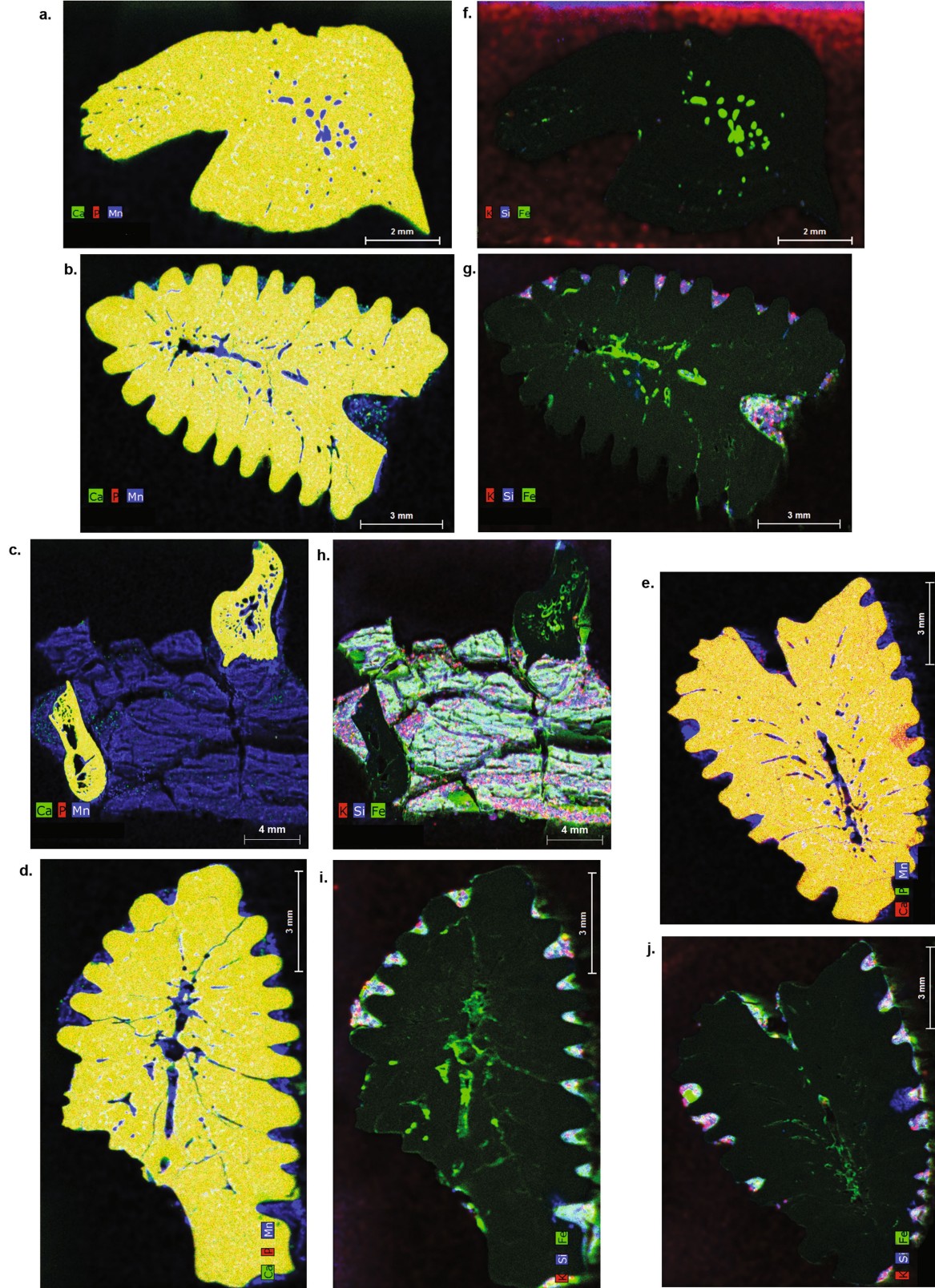

**Extended Data Fig. 8 | Elemental distribution maps of acipenseriform elements from the Tanis locality obtained with micro-X-ray fluorescence.** **a**, Ca, P, and Mn distribution in paddlefish dentary VUA.GG.2017.X-2724. **b**, Ca, P, and Mn distribution in sturgeon pectoral fin spine VUA.GG.2017.MDX-3. **c**, Ca, P, and Mn distribution in paddlefish dentaries VUA.GG.2017.X-2733A, VUA.GG.2017.X-2733B, and the surrounding sediment matrix. **d**, Ca, P, and Mn distribution in sturgeon pectoral fin spine VUA.GG.2017.X-2743M. **e**, Ca, P, and Mn distribution in sturgeon pectoral fin spine VUA.GG.2017.X-2744M. **f**, K, Si, and Fe distribution in paddlefish dentary VUA.GG.2017.X-2724. **g**, K, Si, and Fe distribution in sturgeon pectoral fin spine VUA.GG.2017.MDX-3. **h**, K, Si, and Fe distribution in paddlefish dentaries VUA.GG.2017.X-2733A, VUA.GG.2017.X-2733B and the surrounding sediment matrix. **i**, K, Si, and Fe distribution in sturgeon pectoral fin spine VUA.GG.2017.X-2743M. **j**, K, Si, and Fe distribution in sturgeon pectoral fin spine VUA.GG.2017.X-2744M.

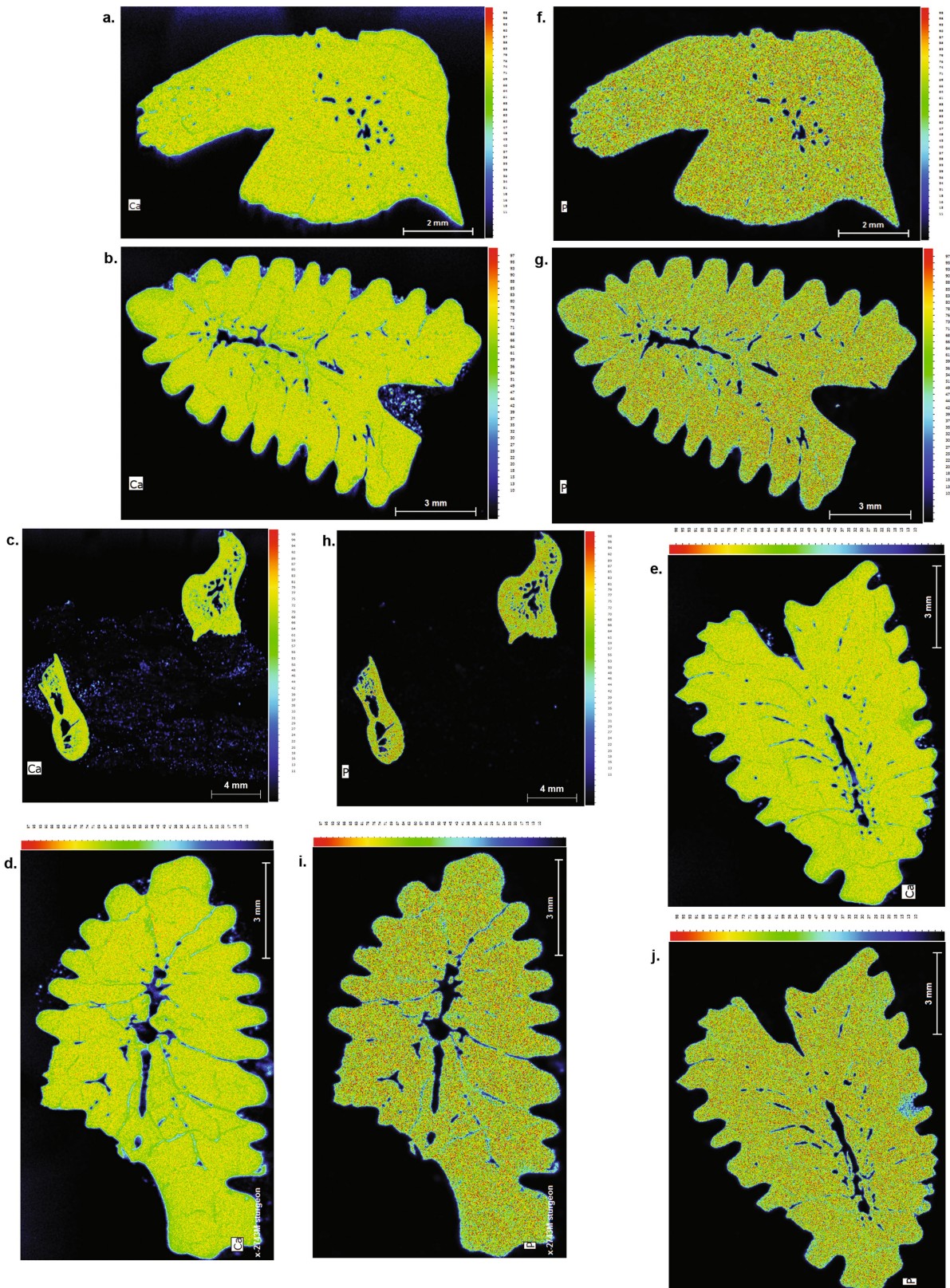

**Extended Data Fig. 9 | Elemental heat maps of acipenseriform elements from the Tanis locality, obtained with micro-X-ray fluorescence, showing homogenous distributions of Ca and P. a**, Ca heat map of paddlefish dentary X-2723. **b**, Ca heat map of sturgeon pectoral fin spine VUA.GG.2017.MDX-3. **c**, Ca heat map of paddlefish dentaries VUA.GG.2017.X-2733A, VUA. GG.2017.X-2733B, and the surrounding sediment matrix. **d**, Ca heat map of sturgeon pectoral fin spine VUA.GG.2017.X-2743M. **e**, Ca heat map of sturgeon pectoral fin spine VUA.GG.2017.X-2744M. **f**, P heat map of paddlefish dentary VUA.GG.2017.X-2724. **g**, P heat map of sturgeon pectoral fin spine VUA. GG.2017.MDX-3. **h**, P heat map of paddlefish dentaries VUA.GG.2017.X-2733A, VUA.GG.2017.X-2733B, and the surrounding sediment matrix. **i**, P heat map of sturgeon pectoral fin spine VUA.GG.2017.X-2743M. **j**, P heat map of sturgeon pectoral fin spine VUA.GG.2017.X-2744M. Red and blue indicate higher, respectively lower abundance of Ca (**a**, **b**, **c**, **d**, and **e**) and P (**f**, **g**, **h**, **i**, and **j**).

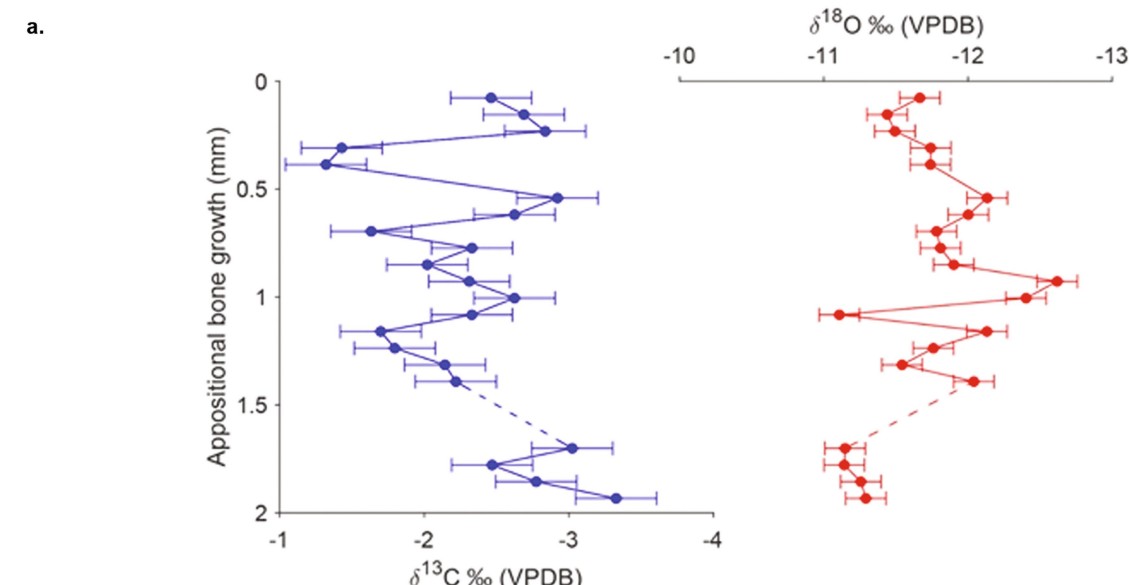

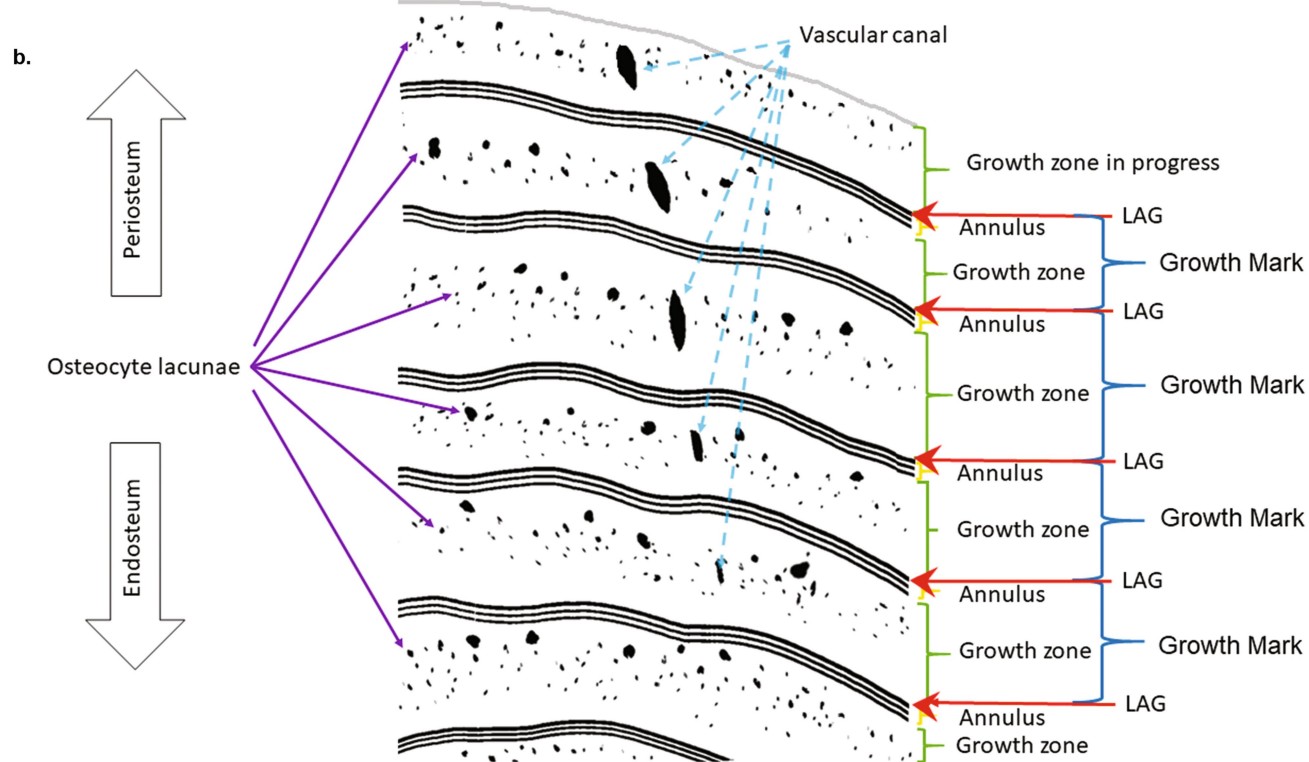

**Extended Data Fig. 10 | $\delta^{13}C_{sc}$ and $\delta^{18}O_{sc}$ data of structurally-bound carbonate across the growth record in paddlefish dentary VUA.GG.2017.X-2724 and osteohistological diagram of paddlefish dentary VUA.GG.2017.X-2724.**
**a**, Incremental records of the carbon ($\delta^{13}C_{sc}$; left) and oxygen ($\delta^{18}O_{sc}$; right) isotopic composition of structurally-bound carbonate in the apatitic matrix of VUA.GG.2017.X-2724 with uncertainty limits (1 s.d.). The seasonal cyclicity of $\delta^{13}C_{sc}$ that reflects alternations in seasonal food availability is, to a lesser extent, also expressed in the $\delta^{18}O_{sc}$ record. **b**, Schematic diagram of paddlefish dentary VUA.GG.2017.X-2724 with osteohistological features indicated. In the acipenseriform dermal bones examined in this study, annual growth cyclicity can be traced through Growth Marks (GMs). A GM spans a single growth cycle that typically lasts one year and can be divided into a zone, an annulus, and a LAG[20,52]. The zone is deposited during a period of relative rapid growth in the active or favourable growth season[20]. The annulus is subsequently formed when growth slows down towards the end of the growth season[20]. Finally, a LAG forms when growth periodically ceases until the next growth season starts and a new zone is deposited[20].

# Reporting Summary

## Statistics

For all statistical analyses, confirm that the following items are present in the figure legend, table legend, main text, or Methods section.

| n/a | Confirmed | |
|---|---|---|
| ☐ | ☒ | The exact sample size (*n*) for each experimental group/condition, given as a discrete number and unit of measurement |
| ☐ | ☒ | A statement on whether measurements were taken from distinct samples or whether the same sample was measured repeatedly |
| ☒ | ☐ | The statistical test(s) used AND whether they are one- or two-sided<br>*Only common tests should be described solely by name; describe more complex techniques in the Methods section.* |
| ☒ | ☐ | A description of all covariates tested |
| ☒ | ☐ | A description of any assumptions or corrections, such as tests of normality and adjustment for multiple comparisons |
| ☐ | ☒ | A full description of the statistical parameters including central tendency (e.g. means) or other basic estimates (e.g. regression coefficient) AND variation (e.g. standard deviation) or associated estimates of uncertainty (e.g. confidence intervals) |
| ☒ | ☐ | For null hypothesis testing, the test statistic (e.g. *F*, *t*, *r*) with confidence intervals, effect sizes, degrees of freedom and *P* value noted<br>*Give P values as exact values whenever suitable.* |
| ☒ | ☐ | For Bayesian analysis, information on the choice of priors and Markov chain Monte Carlo settings |
| ☒ | ☐ | For hierarchical and complex designs, identification of the appropriate level for tests and full reporting of outcomes |
| ☒ | ☐ | Estimates of effect sizes (e.g. Cohen's *d*, Pearson's *r*), indicating how they were calculated |

*Our web collection on statistics for biologists contains articles on many of the points above.*

## Software and code

Policy information about availability of computer code

| Data collection | Inhouse ESRF MATLAB (version 2017a) scripts developed for vertical concatanation, conversion, and ring artefact corrections. |
|---|---|
| Data analysis | ESRF software PyHST2, VGStudio Max 3.2 |

For manuscripts utilizing custom algorithms or software that are central to the research but not yet described in published literature, software must be made available to editors and reviewers. We strongly encourage code deposition in a community repository (e.g. GitHub). See the Nature Portfolio guidelines for submitting code & software for further information.

## Data

Policy information about availability of data

All manuscripts must include a data availability statement. This statement should provide the following information, where applicable:
- Accession codes, unique identifiers, or web links for publicly available datasets
- A description of any restrictions on data availability
- For clinical datasets or third party data, please ensure that the statement adheres to our policy

All scanning data are available at https://doi.org/10.5281/zenodo.5776294 and the http://paleo.esrf.eu database.

March 2021

# Field-specific reporting

Please select the one below that is the best fit for your research. If you are not sure, read the appropriate sections before making your selection.

☐ Life sciences  ☐ Behavioural & social sciences  ☒ Ecological, evolutionary & environmental sciences

For a reference copy of the document with all sections, see nature.com/documents/nr-reporting-summary-flat.pdf

# Ecological, evolutionary & environmental sciences study design

All studies must disclose on these points even when the disclosure is negative.

| | |
|---|---|
| Study description | Osteohistology of six fossil fishes, that died on the day of Chicxulub impact, representing two acipenserifom taxa is combined with isotope analyses of one of these specimens to reveal the seasonality of the last years of the Mesozoic and the season of the impact. Another specimen of acipenseriform fish was furthermore scanned to confirm that these fishes had died because of the accumulation of impact spherules into the gill region at the time of death. |
| Research sample | Seven acipenseriform fishes of which 3 sturgon pectoral fin spines and 3 paddlefish dentaries and 1 partial paddlefish that was scanned nondestructively. The 3 sturgeon pectoral fin spines and 3 paddlefish dentaries were made available to us by the Palm Beach Museum of Natural History  following fieldwork in august 2017 and the partial paddlefish was later made available by the Palm Beach Museum of Natural History for synchrotron scanning. The sample is meant to represent accipenseriform fishes of North America during the latest Cretaceous. |
| Sampling strategy | Four out of six of the samples were excavated from the sediment matrix. These included all sturgeon pectoral fin spines (VUA.GG.2017.X-2743M, VUA.GG.2017.X-2744M, and VUA.GG.2017.MDX-3) and one of the paddlefish dentaries (VUA.GG.2017.X-2724). Paddlefish dentaries VUA.GG.2017.X-2733A and VUA.GG.2017.X-2733B were fractured upon discovery. To avoid further damage, the specimens were embedded in epoxy resin prior to thin sectioning. All specimens were cut with a diamond saw and polished to obtain microscopic thin sections (~50 μm thick) and thick sections for micro milling (~200 μm thick). One partial paddlefish (FAU.DGS.ND.161.4559.T), provided by the Palm Beach Museum of Natural History was only scanned nondestructively |
| Data collection | Fragments of the paddlefish and sturgeon samples that remained after thin sectioning were analysed with Micro X-ray Fluorescence (μXRF) by M.A.D.During and K.H.W. Stein. High-resolution elemental mapping was conducted using a Bruker M4 Tornado 2D spectrometer at 50 kV and 600μA, without a filter, and at an acquisition rate of 20 μm/5 ms at the Vrije Universiteit Brussel (VUB). |
| | The growth increments were sampled in the thick sections by M.A.D.During (~200 μm thick) at the highest possible accuracy using a Micromill (Merkantek). Drill transects were assigned in the accompanying software and after each individual sample was collected, the drill bit was cleaned with ethanol. Not all thick-sections were suitable for micromilling. |
| | Micromilled hydroxyapatite samples of specimen VUA.GG.2017.X-2724 weighing ~50 μg were placed in Exetainer vials (Labco, Lampeter, UK) and flushed with purified helium gas by M.A.D. During. Orthophosphoric acid was subsequently added by S. Verdegaal-Warmerdam and allowed to react for 24 hours at 45°C. VUA.GG.2017.MDX-3 was routinely analysed with a Thermo Finnigan Deltaplus mass spectrometer connected to a Thermo Finnigan GasBench II at the Earth Sciences Stable Isotope Laboratory (Vrije Universiteit, Amsterdam) by S.Verdegaal-Warmerdam. However, the amount of $CO_2$ generated was found to be too small to permit reliable isotopic determinations. To alleviate this, the GasBench was subsequently interfaced with a cold trap by S. Verdegaal-Warmerdam and J.(H)J.L.Van der Lubbe, where the $CO_2$ was frozen with liquid nitrogen during a 2-minute period. After trapping for 2 minutes, an accurate single-pulse measurement was performed, for each of the samples and standards. |
| | Synchrotron data acquisition took place in May 2018 on Beamline BM05 at the European Synchrotron Radiation Facility, Grenoble, France by M.A.D. During, D.F.A.E. Voeten, C. Berruyer & P. Tafforeau. FAU.DGS.ND.161.4559.T was scanned at an average energy of 132 keV using the white beam of BM05 filtered with 0.4 mm of Mo and 9 mm of Cu. The detector was composed of a 2-mm-thick LuAG:Ce scintillator optically coupled to a PCO edge 4.2 CLHS sCMOS camera. The resulting voxel size was 43.5 μm. In order to obtain sufficient propagation phase contrast, the distance between the sample and the detector was set at 5 m. A total of 205 scans, each consisting of 5000 projections taken at 7 ms intervals, were performed with a vertical displacement of 1.4 mm at a vertical field of view of 2.8 mm to ensure a double scan of the complete samples. Scans were performed in half-acquisition mode to enlarge the lateral field of view. The volume was reconstructed using single-distance phase retrieval algorithm coupled with filtered back projection as implemented in the ESRF software PyHST2. Vertical concatenation, 16-bit conversion, and ring artefact corrections were performed using MATLAB scripts developed in-house. The gill region and impact spherules were subsequently scanned at a voxel size of 13.67 μm (filters: 0.4 mm of Mo and 6 mm of Cu, scintillator: LuAG:Ce, 500 μm thick, detected energy: 166 keV, propagation distance: 2.5 m). The samples were scanned in half-acquisition mode in two columns of 77 scans, each consisting of 4998 projections with exposure times of 0.05 s, that were laterally concatenated after reconstruction. |
| | Finally, samples (VUA.GG.2017.X-2724) from the paddlefish dentaries and (VUA.GG.2017.MDX-3, VUA.GG.2017.X-2743M and VUA.GG.2017.X-2744M) sturgeon pectoral fin spines were scanned at 4.35 μm voxel size for osteohistological analysis54 (filters: 3.5 mm of Al plus 11 bars Al with a diameter of 5 mm, scintillator: LuAG:Ce scintillator, 500 μm thick, detected energy: 92 keV, propagation distance: 1.5 m). The samples were scanned in half-acquisition mode in one single column of 22 scans, each consisting of 4998 projections with exposure times of 60 ms. |
| | Digital 3D extraction of the bones and impact spherules was performed in VGStudio MAX 3.2 (Volume Graphics, Heidelberg, Germany) by M.A.D. During and D.F.A.E. Voeten. VGStudio MAX 3.2 furthermore enabled creation of virtual thick sections of the osteohistological samples through the 'thick slab-mode', which captures the maximum, average, or minimum, grey-level values along |

the desired field depth. Virtual thick sections were obtained from the average grey-level values at a thickness of 100 µm following optimal 3D alignment of the annuli and lines of arrested growth (LAGs). Additional virtual thick sections were created from the minimum grey-level values at a thickness of 200 µm to best resolve the sizes and distributions of osteocyte lacunae.

| | |
|---|---|
| Timing and spatial scale | Specimens were obtained from the field in August (1-18) 2017 over an approximate distance of 2 square meters (Specimens were shipped with a delay due to hurricane season).<br>Thin and thick sections were cut on December 5, 2017.<br>X-Ray Fluorescence took place on March 13, 2018.<br>Stable Isotope analyses took place after 2 months of micromilling, without the cold trap on May 28, 2018 and with the cold trap in June 19, 2018.<br>Synchrotron scanning took place in May (3-5), 2018 |
| Data exclusions | Oxygen and Carbon isotopic data for sturgeon pectoral fin spine VUA.GG.2017.MDX-3 were excluded due to the unreliability of the data. The amplitude for the measurements were deemed too small to offer reliable results as a consequence of the small sample size. The incremental micromill sample lines 1-5, 10-12 and 25 for paddlefish dentary VUA.GG.2017.X-2724 did not retrieve sufficient material for isotopic analyses (<2 V) and/or the atmospheric contamination as is indicated by the presence of a nitrogen peak, which led to their exclusion. |
| Reproducibility | For the isotopic analyses, the stability of the mass spectrometer was assured by the isotopic analysis of six reference gas peaks preceding each sample and standard measurement. This so-called monitor gas is routinely calibrated with carbonate standards with internationally accepted values. For linearity corrections, the inter-laboratorial apatitic standard (Ag-lox) has been measured four times within the sample run. The sample sizes that were obtained from specimen VUA.GG.2017.X-2724 do not allow for replicates. The analytical procedure and possible limitations of which are described in detail in the methods section. |
| Randomization | n.a. due to the nature of the available fossil material. |
| Blinding | n.a. due to the nature of the available fossil material. |

Did the study involve field work? ☒ Yes ☐ No

## Field work, collection and transport

| | |
|---|---|
| Field conditions | Conditions in August 2017 varied from extremely hot and dry (~35 degrees Celsius) to extemely wet and roughly 20 degrees Celsius for 3 days, during which we did not excavate any material as we risked getting stuck in the mud. |
| Location | Tanis North Dakota: 46.031403"N, -103.796603"W |
| Access & import/export | Access was permitted via the Palm Beach Museum of Natural history and all specimen transactions as well.<br>The studied specimens were excavated at the Tanis site. Application for off-site shipment has been granted under number X24.4.T of access to research site. Application #: BV60717 |
| Disturbance | Nothing was touched unless it was taken for study, no organisms (wild or agricultural were confronted or hurt) |

# Reporting for specific materials, systems and methods

We require information from authors about some types of materials, experimental systems and methods used in many studies. Here, indicate whether each material, system or method listed is relevant to your study. If you are not sure if a list item applies to your research, read the appropriate section before selecting a response.

## Materials & experimental systems

| n/a | Involved in the study |
|---|---|
| ☒ ☐ | Antibodies |
| ☒ ☐ | Eukaryotic cell lines |
| ☐ ☒ | Palaeontology and archaeology |
| ☒ ☐ | Animals and other organisms |
| ☒ ☐ | Human research participants |
| ☒ ☐ | Clinical data |
| ☒ ☐ | Dual use research of concern |

## Methods

| n/a | Involved in the study |
|---|---|
| ☒ ☐ | ChIP-seq |
| ☒ ☐ | Flow cytometry |
| ☒ ☐ | MRI-based neuroimaging |

## Palaeontology and Archaeology

| | |
|---|---|
| Specimen provenance | Tanis, North Dakota, United States of America: Site: PBMNH.ND.X17.54 Tanis. Access was permitted via the Palm Beach Museum of Natural history and all specimen transactions as well. Application for off-site shipment has been granted under number X24.4.T of access to research site. Application number: BV60717 |

Specimen deposition | All specimens are available at the Palm Beach Museum of Natural History and the VU Amsterdam

Dating methods | No dates are provided

☒ Tick this box to confirm that the raw and calibrated dates are available in the paper or in Supplementary Information.

Ethics oversight | n.a. all specimens are fossil.

Note that full information on the approval of the study protocol must also be provided in the manuscript.

