## [Peer Review File · Nature]

Manuscript Title: The Mesozoic terminated in boreal spring

Editorial Notes:

Redactions – Third Party Material

Reviewer Comments & Author Rebuttals

Reviewer Reports on the Initial Version:

Referee #1 (Remarks to the Author):

Key results and significance:

This article presents results from exceptional specimens that should pique the curiosity of any palaeontologist: fishes that died directly from the asteroid impact coinciding with the K-Pg extinction event. The main premise of the work is a clever use of bone histology to decipher the life history of these animals, providing clues towards the seasonality of this extinction event.

As I am more a specialist of systematics, anatomy and bone histology, I will refrain providing comments on the significance of this to the community specifically studying mass extinctions. However, as a palaeo-ichthyologist that also uses bone histology, I'm elated to see fossil ray-finned fishes to be used to address such general questions in palaeontology, and hope this will generate further interest in the field.

Methodology:

I do not have the technical skills to evaluate the methodology of the study pertaining to X-ray fluorescence elemental mapping and stable isotope analysis. Therefore, my review focuses on the bone histology and PPC-SR μ CT (tomography).

The method to produce thin sections and SR μ CT data is sound, and follows the standards established in the field, often by some of the authors themselves.

Something that I wondered a bit while reading the manuscript is what was the rationale for using SR μ CT for this particular study. Since physical thin sections are also available (and visible in Fig. 2 and the Supplements), what information on bone histology do CT data bring that thin sections would not? I think this needs to be clearer in the manuscript, as it would reinforce the significance of the study for using cutting-edge imaging methods.

The paper would also benefit from a short statement on the usefulness and adequateness of SR μ CT data for bone histology: this has been shown repeatedly by co-authors Sophie Sanchez and Paul Tafforeau (Sanchez et al. 2012, *Microscopy and Microanalysis*; 2014, *Proc. Roy. Soc. B*; 2016, *Nature*, and so on). A recent study, specifically on actinopterygians, also shows that these data can be used (amongst others) in the context of cyclical growth (Davesne et al. 2020, *J. Evol. Biol.*).

Appropriate use of statistics and treatment of uncertainties: -

Conclusions:

The conclusions appear robust to me, especially since they are informed by different approaches (histology of bone growth and carbon isotopes).

The explanations on annual growth are clear and adequate. However, I feel that the two following sentences (from L 103 to L 107) do not explain sufficiently the link between the observations and the conclusions. Why having a lower osteocyte lacunar density means that bone growth was not slowing down? Was this quantified in some way, and can it be seen clearly from the sections and/or μ CT data? Figure 2 and its caption do not provide much more information on that matter as it is. I would recommend providing more details and context.

Suggested improvements:

Another point that was insufficiently exposed in the manuscript is the identification and systematic attribution of the specimens.

No photograph of the fossils is available in the current manuscript, or in the Supplements. In the original article on the fossil site (DePalma et al. 2019, PNAS) none of the figures showed the fish fossils in enough details to identify them, even broadly. Only in the supplements of the DePalma et al 2019 paper, was I able to find a figure (Fig. S22) that showed enough detail to confirm that indeed, these are acipenseriforms.

While I understand that length constraints apply here, I think a photograph of the specimens, and particularly, of the sampled bone elements in context (i.e., before having been sampled from the entire fish fossil), would help a lot in making this clearer. Since the conclusions of the article partially rely on the identity of the specimens as acipenseriforms, I think this is an important point that needs to be addressed (at the very least in the Supplements).

On a related note, I feel Figure 1 is a bit hard to interpret. The bones that were segmented are rather dislocated so it is difficult to understand where are the jaws, opercles, cranial roof, etc. A bit more legends could help, or an interpretative drawing, or some drawing that includes an approximate silhouette of the skull. Even with the eye symbol, I think it is difficult for the reader to understand what they are looking at.

Coming back to the identity of the fossils, I would have liked more information on their identification besides pictures of the specimens. This can be put in the Supplements, alongside a brief sentence in the main text. One particular detail struck my attention: the thin sections MDX-3, 2743M and X-2744M, supposedly of acipenserid (sturgeon) spines, look peculiar to me. They show radial structures, in addition to osteocyte lacunae and canaliculi, that look a lot like canaliculi of Williamson. These structures are only known so far in the bone and scales of fossil and extant holosteans (particularly lepisosteids) and some stem-group teleosts from the Mesozoic (e.g. Sire & Meunier 1994, *Acta Zoologica*; Meunier & Brito 2004, *Cybium*). They are not supposed to exist in modern sturgeons, as far as I know (i.e. Meunier, François & Castanet 1978, *Bull. Soc. Zool. Fr.*). I am therefore particularly interested in the identity of these spines. Lepisosteids, known for their canaliculi of Williamson, are abundant in the Late Cretaceous of North America, but do not have such spines and have a very different morphology in general, so I do not think this is an error of attribution but once again it is impossible to be sure without a picture of the complete fossil.

Small typos:

L23 and L95: Please verify that the use of "perichondrial bone" is correct. "Perichondral" occurs more often.

L31: "caused by the Chicxulub impact"

L73: perimortem (not perimortum).

References : Appropriate.

Clarity and context: Appropriate.

Many thanks for inviting me to review this very interesting manuscript!

Referee #2 (Remarks to the Author):

Smit et al. have produced a solid study pinpointing the timing of the end-Cretaceous impact based on the growth records of fish which they demonstrate to have died instantaneously at impact. Their evidence that the impact occurred in Spring is strong, and their discussion of the explanatory power of this timing for extinction selectivity snaps a lot of disparate evidence into place. This study creates a full picture of the kill mechanism of the end-Cretaceous mass extinction, will help reconcile conflicting patterns from different regions and clades, and sets an agenda for future paleobiogeographic research and mass extinction selectivity work (e.g. focusing on life histories; it's too bad most other mass extinctions did not start on a single day!). Looking at histology to determine mass extinction causes is an uncommon and unexpected approach, and the pay off is great. In sum, this paper is nothing short of groundbreaking and represents a major discovery worthy of publication in Nature. I recommend publication as is, and I am not sure I ever done that before!

Referee #3 (Remarks to the Author):

Review of During et al: "Bone histology of acipenseriform fishes reveals seasonality during the final years of the Mesozoic"

During et al present a unique and novel dataset of exceptionally preserved fish fossils that were likely killed by the Chicxulub impact at the end of the Mesozoic Era. The fossils come from the Tanis locality in North Dakota, USA, a site interpreted as a seiche deposit caused by the bolide impact that preserves the minutes to hours after the impact occurred. Their findings, using a combination of histological and stable carbon isotope data on fin spines and dentaries from sturgeon and paddlefish preserved at the site, suggest that the bolide impact occurred in the boreal spring. Their methodology is unique, taking advantage of the Tanis locality's precise preservational history, and using seasonal and annual fish growth patterns to reconstruct the season in which the fish preserved at Tanis perished. The conclusion they reach is well in line with work by Wolfe and others, which used paleobotanical evidence to suggest a June (e.g. late spring or early summer) impact timing. It is encouraging that these two methods, using entirely independent methods, study systems, and organismal groups, working decades apart, came to very similar conclusions. Their study provides additional evidence for the seasonality of the bolide impact, which has implications for selective extinctions and survival, which the authors discuss briefly at the end of the manuscript. Overall, the manuscript is easy to read, although contains considerable extraneous information, a number of less-than-relevant references, and many missing references that are more appropriate to their discussion.

The methods, data, and overall results appear relatively robust (the carbon isotopes as an indicator for seasonality based on prey field is particularly intriguing), however the manuscript itself, and the context in which the authors frame this dataset, are not yet ready for publication. Some general comments:

1. The authors have demonstrated that one fish preserved in the deposit died with impact

spherules in its gill rakers, suggesting that it was smothered rapidly as the impact spherules began to rain down. However, this single specimen is then used to extrapolate that all fish preserved in the deposit perished simultaneously – perhaps not an incorrect assumption, and one in which the authors give considerable space to discussing, however in the end, they present data from only one specimen, and this is not conclusive, and the specimens that they use for their analyses do not have this set of properties.

2. The authors include only two species of fish (sturgeon spines and paddlefish dentaries) in their discussions – and just few specimens of each. The data they present in Figure 2 is only from one specimen, and includes only stable isotope and images from that single specimen. Where are the additional specimens and data? It does not appear to be in the supplemental data and figures either.

3. The authors need to include more than the single specimen in their figures and analyses – it appears that not all specimens have all data types, due to preservation, but they discuss in detail fin spines in the text, and do not show images or isotope data. As it stands, their argument is based on a single specimen and that is not sufficient data.

4. To this end, Figure 2 should include a panel showing density (light/dark) along the growth lines for this specimen, demonstrating seasonality in growth in addition to the carbon isotopes. Figure 2 should also include similar data and images of the other specimens included in their analyses.

5. The reasoning of seasonality, based on both bone density and carbon isotopes, which vary based on the seasonal prey field, are in alignment with each other based on the text, but are not shown effectively in the figures.

6. Why were there no oxygen isotope measurements made on the micro-drilled material? This seems an obvious and straightforward way to get at seasonal water temperatures.

7. Finally, I question the authors conclusion of “spring” as opposed to summer or even fall – The authors show that there is a clear seasonal cycle in carbon isotopes in figure 2a, however the length of that seasonal cycle is unclear, and it appears that the edge of the bone (interpreted as time of death) is in the middle of the “growth” season, not at the very beginning or very end. Since they present the cyclical data only from a single specimen, and the cycles are varied in magnitude and length, it is not sufficiently clear what this would be showing if additional specimens were included. While the authors say that this is “spring”, I think that it could be anywhere in the “spring to fall” region based on the data that they present, although likely was closer to late spring or early summer – well in line with Wolfe (1991)’s estimate of “June”.

Some line by line comments:

Line 17 – need a citation here

Line 21 – “... leaving the season of the impact unconstrained” – except for paleobotanical evidence, which should be mentioned here

Line 25 – “These fishes ultimately perished in boreal spring” is repetitive

Lines 28/29 – There is not evidence in the present manuscript that the boreal spring impact “significantly influenced selective biotic survival across the KPg boundary”. At best, it is a hypothesis that is suggested by the authors in the discussion, and should be couched as such in the abstract.

Paragraph starting on line 31: This paragraph feels somewhat out of place and could be mostly cut, particularly given the context already stated in the abstract. The first sentence about the debate of causal mechanisms is completely unnecessary.

Line 38 – the citations here are backwards (should be “2, 7” not “7, 2”)

Line 52/53 – this sentence is unnecessary, and R. DePalma is mentioned in the acknowledgements. Pers comm not appropriate to cite in text

Lines 54-63 – are there any localities beyond Hell Creek that can provide a more nuanced and localized view of seasonality for Tanis? Seasonality can vary greatly a long a latitudinal band depending on proximity to the coast, atmospheric patterns, etc., and the references used here to “reconstruct” seasonal temperature estimates are not precise temporally or geographically.

Lines 87-90 – it is possible to have 3D preservation and also have significant replacement of minerals in the skeletal elements. The fact that there is 3D preservation does not provide evidence for lack of chemical changes.

Lines 93/94 *and* Lines 113-115 – there are often multiple peaks of phytoplankton and zooplankton blooms in temperate systems, spring and fall. Please discuss how you differentiated these, if at all, and why you interpret “spring” as the season rather than summer or fall.
Line 122 – these references do not seem to support this sentence
Final paragraph – this is a lot of interesting speculation, and is missing citations to relevant literature that has discussed these geographic and ecosystem selectivity regimes.
Line 307 – please include specific permanent URLs to the specimens

Figure 1 – this did not translate well to black and white/printing, although the googly eye placement made me smile. I appreciate the necessity of this figure, and panel B is particularly interesting and well done.

Figure 2 – please see comments above. This figure is a good start, but is missing key information.

Comments on supplementary information: many of the scale bars are extremely hard to read because the text is too small (e.g. pages 3, 4, 6, 7, 9, 10, 12, 13, 15, 16, 18, 19)
Page 39 (SI) has some extra text on the image

Referee #4 (Remarks to the Author):

Dear editor,

The manuscript by Doring et al. presents a unique bone histological record from several fossil fishes from the spectacular Tanis Cretaceous-Paleogene boundary seiche deposit. Based on impact spherules lodged in their gills, it can be assumed that these fish died on the day of the Chicxulub asteroid impact. The bone histology and stable carbon isotope record of one of these fishes suggests that the death of the studied animal occurred during boreal spring time, providing a tentative constrain on the season in which the Chicxulub impact occurred.

The study is highly original and absolutely significant, both for the Cretaceous-Paleogene boundary impact community as well as for a wider audience. The approach is valid, data of good quality, with appropriate treatment of uncertainties, presentation is simple but clear. Perhaps the main figure could use some additional labels, to guide the less-informed reader a bit better (see attached list of comments and suggestions).

The presented data support the main conclusion of the manuscript, that is that the Chicxulub impact occurred in boreal spring. Nevertheless, with only one stable isotope record presented, the robustness remains difficult to assess. According to the Materials & Methods section, two specimens were measured for isotopes, while only one is provided in the manuscript. If both records, both sampled up to the outermost growth increment, would be presented, this would considerably strengthen the case. While here and there the choice of references might not be optimal (see attached list of comments and suggestions), in general the authors appropriately credit previous work.

All in all, the manuscript is clearly written, the abstract clear and appropriate and the introduction and conclusions are all easy to follow. Even though there are several minor issues with the (re)presentation of the data, the high quality of the research, its broad significance and major potential impact lead me to recommend this manuscript to be accepted with revisions.

Author Rebuttals to Initial Comments:

Referee #1 (Remarks to the Author):

Key results and significance:

This article presents results from exceptional specimens that should pique the curiosity of any palaeontologist: fishes that died directly from the asteroid impact coinciding with the K-Pg extinction event. The main premise of the work is a clever use of bone histology to decipher the life history of these animals, providing clues towards the seasonality of this extinction event.

As I am more a specialist of systematics, anatomy and bone histology, I will refrain providing comments on the significance of this to the community specifically studying mass extinctions. However, as a palaeo-ichthyologist that also uses bone histology, I'm elated to see fossil ray-finned fishes to be used to address such general questions in palaeontology, and hope this will generate further interest in the field.

Methodology:

I do not have the technical skills to evaluate the methodology of the study pertaining to X-rayfluorescence elemental mapping and stable isotope analysis. Therefore, my review focuses on the bone histology and PPC-SR μ CT (tomography).

The method to produce thin sections and SR μ CT data is sound, and follows the standardsestablished in the field, often by some of the authors themselves.

Something that I wondered a bit while reading the manuscript is what was the rationale for using SR μ CT for this particular study. Since physical thin sections are also available (and visible in Fig. 2 and the Supplements), what information on bone histology do CT data bringthat thin sections would not? I think this needs to be clearer in the manuscript, as it would reinforce the significance of the study for using cutting-edge imaging methods.

The paper would also benefit from a short statement on the usefulness and adequateness ofSR μ CT data for bone histology: this has been shown repeatedly by co-authors Sophie Sanchez and Paul Tafforeau (Sanchez et al. 2012, Microscopy and Microanalysis; 2014, Proc. Roy. Soc. B; 2016, Nature, and so on). A recent study, specifically on actinopterygians, also shows that these data can be used (amongst others) in the context of cyclical growth (Davesne et al. 2020, J. Evol. Biol.).

Response: As mentioned by Referee #1, several papers have demonstrated the advantages of combining traditional histology with virtual 3D osteohistology. Physical sectioning of bone unlocks direct comparison against abundant published reference materials on microstructuralanalysis revealed by both natural and polarised light, with scalable resolutions. Non-invasive virtual osteohistology perfectly complements these methods by permitting completely three-dimensional characterisation of bone microstructures and granting valuable tools for visualising subtle and hard-to-resolve features in a completely dynamic spatial environment. Combining traditional and virtual

osteohistology provides the best of both techniques in visualising and characterising specific features of interest in fossil bone. We here used this combined approach for optimal visualisation and evaluation of growth marks and spatial as well as size distributions of osteocyte lacunae towards best capturing cyclical growth. We now explain the advantages of using 3D synchrotron visualisation alongside physical slides in lines 78 – 86, supported by the additional references as suggested by Referee #1.

Appropriate use of statistics and treatment of

uncertainties: -Conclusions:

The conclusions appear robust to me, especially since they are informed by different approaches (histology of bone growth and carbon isotopes).

The explanations on annual growth are clear and adequate. However, I feel that the two following sentences (from L 103 to L 107) do not explain sufficiently the link between the observations and the conclusions. Why having a lower osteocyte lacunar density means that bone growth was not slowing down? Was this quantified in some way, and can it be seen clearly from the sections and/or μ CT data? Figure 2 and its caption do not provide much more information on that matter as it is. I would recommend providing more details and context.

Response: We agree with Referee #1 that the referred section should be improved. We, therefore, clarified that osteocyte lacunar density and average size show an annual cyclicity, with both parameters peaking during the climax of the favourable growth season (i.e. summer). Since we record an increasing osteocyte lacunar density and average size during the final year of growth in all six bone records, but also recognise that the peak osteocyte lacunar density and size were not yet achieved, we conclude that growth, therefore, must have ceased after the unfavourable growth season (i.e. winter; a period of no or very limited bone accretion) but well prior to the annual growth climax (i.e. summer). This implies a spring death. The improved clarification is presented in lines 120 – 130 and extended to original Figure 2 (Now Figure 3), where the highest osteocyte lacunar densities have now been shaded green and clarified in the caption.

Suggested improvements:

Another point that was insufficiently exposed in the manuscript is the identification and systematic attribution of the specimens.

No photograph of the fossils is available in the current manuscript, or in the Supplements. In the original article on the fossil site (DePalma et al. 2019, PNAS) none of the figures showed the fish fossils in enough details to identify them, even broadly. Only in the supplements of the DePalma et al 2019 paper, was I able to find a figure (Fig. S22) that showed enough detail to confirm that indeed, these are acipenseriforms.

While I understand that length constraints apply here, I think a photograph of the

specimens, and particularly, of the sampled bone elements in context (i.e., before having been sampled from the entire fish fossil), would help a lot in making this clearer. Since the conclusions of the article partially rely on the identity of the specimens as acipenseriforms, I think this is an important point that needs to be addressed (at the very least in the Supplements).

Response: We agree with Referee #1 that conclusive acipenseriform identification of the studied specimens is crucial for supporting the inferences made, as it offered valuable comparative context against well-understood (modern) reference materials and permitted consideration of potential taxon-specific conditions (e.g. the possibility of an anadromous lifestyle in modern sturgeons). DePalma et al (2019) report that the most abundant “(hundreds or more)” vertebrates at Tanis are paddlefishes and sturgeons. This appendix also reports the fragility of the (sampled) skeletal associations, which, together with the instable nature (due to post-depositional dewatering and shrinkage) of the surrounding matrix, rendered them very challenging to accurately photograph them in the field. To better present the studied skeletal elements, we have now added pictures of two of the three sampled sturgeon fin spines in the field closely associated with the characteristic scutes of the sturgeon specimens they were sourced from. These photographs are included as Supplementary Information 2.1. Additionally, we have added photographs of all specimens prior to sampling, revealing some diagnostic characters, such as the characteristically convoluted exterior morphology of sturgeon fin spines, as in Supplementary Information 2.2. Finally, Supplementary Information 5.5 and Video V1. presents an undoubtable paddlefish dentary in its anatomical context, the morphology of which closely matches those of the sampled paddlefish dentaries in Supplementary Information 2.2.

On a related note, I feel Figure 1 is a bit hard to interpret. The bones that were segmented are rather dislocated so it is difficult to understand where are the jaws, opercles, cranial roof, etc. A bit more legends could help, or an interpretative drawing, or some drawing that includes an approximate silhouette of the skull. Even with the eye symbol, I think it is difficult for the reader to understand what they are looking at.

Response: We have closely followed the suggestions by Referees #1, #3, and #4 to improve the clarity of Figure 1. All the larger morphological attributes of the specimen, such as the pectoral girdle, skull roof, jaw, and gill cover, have now been annotated to guide the reader. Additionally, the referred skeletal representation is now also provided in both lateral views as Supplementary information 5.4, in which all the preserved bones are individually annotated.

Coming back to the identity of the fossils, I would have liked more information on their identification besides pictures of the specimens. This can be put in the Supplements, alongside a brief sentence in the main text. One particular detail struck my attention: the thin sections MDX-3, 2743M and X-2744M, supposedly of acipenserid (sturgeon) spines, look peculiar to me. They show radial structures, in addition to osteocyte lacunae and

canaliculi, that look a lot like canaliculi of Williamson. These structures are only known so far in the bone and scales of fossil and extant holosteans (particularly lepisosteids) and some stem- group teleosts from the Mesozoic (e.g. Sire & Meunier 1994, *Acta Zoologica*; Meunier & Brito 2004, *Cybium*). They are not supposed to exist in modern sturgeons, as far as I know (i.e.

Meunier, François & Castanet 1978, *Bull. Soc. Zool. Fr.*). I am therefore particularly interested in the identity of these spines. Lepisosteids, known for their canaliculi of Williamson, are abundant in the Late Cretaceous of North America, but do not have such spines and have a very different morphology in general, so I do not think this is an error of attribution but once again it is impossible to be sure without a picture of the complete fossil.

Response: We thank referee #1 for sharing these insights and are indeed aware that the studied materials originate from taxa that may deviate slightly from their present-day relatives. We also agree that the identity of the referred sturgeon fin spines is conclusively established, as further corroborated through the close anatomical association between these fin spines and the scutes that are diagnostic for the Tanis sturgeons (Supplementary Information 2.1). Presentation of the sampled elements as Supplementary Information 2.2 enables independent confirmation of the identity of the sturgeon fin spines and paddlefish dentaries. Moreover, the preserved growth records extracted from all confirmed acipenseriform elements consistently attest to a spring death, irrespective of the osteohistological peculiarities of the preserved materials. It should furthermore be noted that, despite a thorough deposit-wide palaeontological survey (DePalma et al., 2019), lepisosteid remains were not encountered at the Tanis locality. This would make a lepisosteid identity of no less than three elements recovered from a near-exclusive acipenseriform association virtually impossible, especially because the conspicuous ganoid scales of gars would have been quite abundant and easily recognisable. Further identification of these fish species is outside the scope of this study. Most importantly, the conclusions, based on 6 specimens spanning both abundantly available acipenseriform species from the Tanis site, of the presented study are robust, as mentioned by Referee #1.

Small typos:

L23 and L95: Please verify that the use of “perichondrial bone” is correct.
“Perichondral” occurs more often.

✓ Corrected

L31: “caused by the Chicxulub impact”

✓ Corrected

L73: *perimortem* (not *perimortum*).

✓ Corrected

References : Appropriate. Clarity and context: Appropriate.

Many thanks for inviting me to review this very interesting manuscript!

We thank Referee #1 for the valuable input and discussion on our manuscript.

Referee #2 (Remarks to the Author):

Smit et al. have produced a solid study pinpointing the timing of the end-Cretaceous impact based on the growth records of fish which they demonstrate to have died instantaneously at impact. Their evidence that the impact occurred in Spring is strong, and their discussion of the explanatory power of this timing for extinction selectivity snaps a lot of disparate evidence into place. This study creates a full picture of the kill mechanism of the end-Cretaceous mass extinction, will help reconcile conflicting patterns from different regions and clades, and sets an agenda for future paleobiogeographic research and mass extinction selectivity work (e.g. focusing on life histories; it's too bad most other mass extinctions did not start on a single day!). Looking at histology to determine mass extinction causes is an uncommon and unexpected approach, and the pay off is great. In sum, this paper is nothing short of groundbreaking and represents a major discovery worthy of publication in Nature. I recommend publication as is, and I am not sure I ever done that before!

We thank Referee #2 for sharing our enthusiasm for this study.

Referee #3 (Remarks to the Author):

Review of During et al: "Bone histology of acipenseriform fishes reveals seasonality during the final years of the Mesozoic"

During et al present a unique and novel dataset of exceptionally preserved fish fossils that were likely killed by the Chicxulub impact at the end of the Mesozoic Era. The fossils come from the Tanis locality in North Dakota, USA, a site interpreted as a seiche deposit caused by the bolide impact that preserves the minutes to hours after the impact occurred. Their findings, using a combination of histological and stable carbon isotope data on fin spines and dentaries from sturgeon and paddlefish preserved at the site, suggest that the bolide impact occurred in the boreal spring. Their methodology is unique, taking advantage of the Tanis locality's precise preservational history, and using seasonal and annual fish growth patterns to reconstruct the season in which the fish preserved at Tanis perished. The conclusion they reach is well in line with work by Wolfe and others, which used paleobotanical evidence to suggest a June (e.g. late spring or early summer) impact timing.

Response: We thank Referee #3 for the clear summary that underlines the rationale behind our study.

We are aware that Wolfe (1991) indeed arrived at a similar conclusion but the argumentation for Wolfe's interpretation has since been questioned with regards to 1) palaeobotanical identification (Nichols 1992; Hicky and McWeeney, 1992), 2) the uniqueness of morphological features, initially attributed to freezing (Hicky and McWeeney, 1992; McIver et al., 1999; Upchurch et al., 2007), and 3) the stratigraphic position of the studied materials relative to the K-Pg event and the hypothesis requiring two bolide impacts (Nichols 1992; McIver 1999).

For details, we refer Referee #3 to the aforementioned references, but a boreal spring impact as inferred by Wolfe (1991) is neither well constrained nor supported by more recent studies. Therefore, Wolfe (1991) might have been right for the wrong reasons. Our study provides the first independent and robust evidence for the seasonal timing of the K-Pg impact.

It is encouraging that these two methods, using entirely independent methods, study systems, and organismal groups, working decades apart, came to very similar conclusions. Their study provides additional evidence for the seasonality of the bolide impact, which has implications for selective extinctions and survival, which the authors discuss briefly at the end of the manuscript. Overall, the manuscript is easy to read, although contains considerable extraneous information, a number of less-than-relevant references, and many missing references that are more appropriate to their discussion.

The methods, data, and overall results appear relatively robust (the carbon isotopes as an indicator for seasonality based on prey field is particularly intriguing), however the manuscript itself, and the context in which the authors frame this dataset, are not yet ready

for publication. Some general comments: The authors have demonstrated that one fish preserved in the deposit died with impact spherules in its gill rakers, suggesting that it was smothered rapidly as the impact spherules began to rain down. However, this single specimen is then used to extrapolate that all fish preserved in the deposit perished simultaneously – perhaps not an incorrect assumption, and one in which the authors give considerable space to discussing, however in the end, they present data from only one specimen, and this is not conclusive, and the specimens that they use for their analyses do not have this set of properties.

Response: We realise that our earlier manuscript drew too much attention to the single fish with impact spherules in its gill rakers that was visualised at the European Synchrotron and failed to make clear that this observation complements those of DePalma et al. (2019).

DePalma et al. (2019) already report that “spherules are concentrated in the gill rakers of more than 50% of acipenseriform fish carcasses within the [Tanis] deposit.” that are “The most abundant (hundreds or more) vertebrates at Tanis...” (see also the response to Referee #1). This publication provides detailed pictures (Fig. 6 and Supplementary Fig. 15 in DePalma et al., 2019) showing impact spherules in gill rakers, as well as a CT scan of another fish skull with impact spherules in its gill rakers (Fig. 6 in DePalma et al., 2019).

Outside gill rakers, spherules were encountered throughout the surrounding event deposit, occasionally at the bottom of their impact funnels (DePalma et al., 2019), but never within the fish carcasses. We have now improved the clarity by a reference to DePalma et al., 2019.

Our detailed SR μ CT data of another specimen than those imaged in DePalma et al. (2019) again reveals that the gill rakers are filled with impact spherules and that the matrix surrounding the three-dimensional body outline contains impact spherules, but that spherules had not entered the fish body during life or hypothetical post-mortem exposure, as they are absent within the body outline.

As elaborated on by DePalma et al. (2019), the preservation of spherules with impact structures *throughout* the deposit attests to its near-instantaneous deposition. This implies that all biota within the deposit perished and were buried at virtually the same moment, as also indicated by the universal orientation of fish carcasses pointing to directional forward and reverse surges (see figure below). Therefore, the six individual fishes that were osteohistologically assessed here can, together with the event deposit they are embedded in, be confidently placed at the same moment “minutes after the impact” (DePalma et al., 2019).

[Figure redacted]

1. *The authors include only two species of fish (sturgeon spines and paddlefish dentaries) in their discussions – and just few specimens of each. The data they present in Figure 2 is only from one specimen, and includes only stable isotope and images from that single specimen. Where are the additional specimens and data? It does not appear to be in the supplemental data and figures either.*

Response: We understand from this question that we have probably not cited our Supplementary Information, which shows the diversity of the data that we collected, with appropriate clarity. To improve this, we have now included a new figure (Figure 2) in the main manuscript presenting the five additional osteohistological thin sections in which the Lines of Arrested Growth (LAGs) are indicated. Furthermore, we have included more references to the osteohistological data of these five other fishes that were analysed alongside the specimen presented in the original original Figure 2 (now Figure 3). The PPC- Sr μ CT data of the five other specimens are included in the Supplementary Information (5.1 and 5.2) and all reveal mutually consistent growth histories that converge on a spring death. Although all fish specimens were sampled for stable isotope analysis, the narrow diameter of the growth bands did not retrieve sufficient material for stable isotope analysis in five of them. Micromilling the growth increments at sufficient resolution was only successful in MDX-3 and X-2724 (see Methods section for details). Coarser sampling resolution in the other specimens towards securing sufficient material for stable isotope analysis was no option, as it would have resulted in averaged sampling across multiple growth bands by mixing material from adjacent zones, annuli, and LAGs.

Initially, we applied a routine measurement protocol on the MDX-3 samples weighing 50 μ g, which were measured multiple times (~9). Nevertheless, after these incremental samples were already digested, the produced amplitudes, proportionally linked to the amount of CO₂, remained too small for reliable isotopic determinations.

For reference, the individual incremental apatitic samples of X-2724 (50 μ g on average) yield low-weight percentages of structural carbonate, which are equivalent to ~9 μ g of CaCO₃. To negotiate the issue of minute sample weights for implementing the conventional protocol, as was encountered for MDX-3, we applied a cold trap for the isotope analysis of specimen X- 2724. Rather than averaging 9 subsequent measurement peaks per CO₂ sample, we now trapped the generated CO₂ of each sample using a cold trap that was subsequently released as a single CO₂ pulse and analysed by the mass spectrometer. This method proved successful in securing sufficient material to obtain a reliable stable isotope record across the preserved growth record. We recognise that this approach was not sufficiently explained in the original submission and now detail the followed protocol in the Methods section under “Stable Isotope Analysis”.

2. The authors need to include more than the single specimen in their figures and analyses –it appears that not all specimens have all data types, due to preservation, but they discuss in detail fin spines in the text, and do not show images or isotope data. As it stands, their argument is based on a single specimen and that is not sufficient data.

Response: We agree that the six mutually consistent records, spanning no less than three sturgeon fin spines and three paddlefish dentaries, are best all presented in the main manuscript. However, because we wish to avoid clutter in the original Figure 2, we have now created a new Figure 2 that consists of the thin sections of the specimens not depicted in original Figure 2 (now Figure 3). We also more specifically guide the reader to

Supplementary Information 3, 5.1, and 5.2, which present the complete osteohistological archive of the other five fishes. This reference is now included twice in the relevant portion of the main text as well as in the caption of Figure 3.

Referee #3 is correct in recognising that, despite all six studied fishes yielding unambiguous growth records, only one specimen (X-2724) produced a reliable incremental isotope record as well. Nevertheless, our “argument” equally draws from the growth records of the five other fishes, which are fully (and mutually) consistent with those of X-2724. Moreover, it is important to understand that the stable isotope record provides additional and independent corroboration for our interpretation of the appositional bone growth records rather than an imperative validation of these bone growth signals.

3. To this end, Figure 2 should include a panel showing density (light/dark) along the growthlines for this specimen, demonstrating seasonality in growth in addition to the carbon isotopes. Figure 2 should also include similar data and images of the other specimens included in their analyses.

Response: We appreciate the suggestion to better indicate seasonality in Figure 3 (original figure 2). We have therefore colour coded the backdrop of Figure 3a to improve contextualisation of the fluctuating carbon isotope curve during appositional growth. Figure 3a now communicates that elevated carbon isotope values, indicating more feeding, coincide with stages of accelerated growth in the warmer seasons. Conversely, lower carbon isotope values, indicating less feeding, coincide with reductions in growth rate in the colder seasons. In addition, we have improved the caption of Figure 3 to better explain the data and interpretations. Furthermore, we have shaded the osteocyte lacunae in green in Figure 3c, indicating zones of bone growth. We do not believe that including all panels showing the complete osteohistological data set of the five other fishes equivalent to Figure 3b-d, would improve the clarity of the paper. Nevertheless, we have improved the structure and legibility of the supplementary information and added strategically placed references to these osteohistological archives in the main text. Now, the reader is referred to the additional osteohistological records in both the manuscript text and the caption of Figure 3.

4. The reasoning of seasonality, based on both bone density and carbon isotopes, which vary based on the seasonal prey field, are in alignment with each other based on the text, but are not shown effectively in the figures.

Response: This has been resolved simultaneously with comment 4 of Referee #3. We have improved the layout of original Figure 2 (now Figure 3) and the associated caption to better highlight the correspondence of the seasonal carbon isotope cycles (i.e. annual feeding patterns) and bone microstructural features indicative of growth (i.e. bone zonation as well as osteocyte lacunar densities and sizes). This linked isotopic and growth record of X-2724, combined with the independent growth records of the five other

specimens, unambiguously resolves seasonality during the final years of the Cretaceous up to a simultaneous death during the “first minutes” after the Chicxulub impact (DePalma et al., 2019).

5. Why were there no oxygen isotope measurements made on the micro-drilled material? This seems an obvious and straightforward way to get at seasonal water temperatures.

Response: We thank Referee #3 for mentioning the oxygen isotope record. We indeed successfully obtained one oxygen isotopic record from the structurally-bound carbonate of X-2724 in conjunction with the carbon isotopic record. These oxygen isotope data are now included as Supplementary Information 6. However, the interpretation of incremental oxygen isotopic composition of structurally-bound carbonate in fish apatite is much more challenging to reliably interpret as a palaeo-environmental proxy and not imperative for consolidating our arguments, and was therefore not included in the initial submission. To our best knowledge, no oxygen isotopic temperature equation exists for structurally bound carbonate in fish apatites. The oxygen isotope composition of phosphate in apatites is significantly correlated with both temperature and the oxygen isotope composition of the water, very much like calcium carbonates of shells and foraminifera (Pucéat et al. 2010, Kolodny et al. 1983). For mammals, a strong correlation between the oxygen isotope composition of phosphate and structural carbonate in apatite tissues has been demonstrated (Pellegrini et al. 2011). However, this relationship is not well-constrained for fishes (Venneman et al., 2001), which renders temperature reconstructions based on structural carbonate obtained from phosphates in fish apatite presently impossible.

Considering these limitations, the oxygen isotopic data does still provide some broader but intertwined constraints on the changes in oxygen isotopic composition of the water and temperature of the water in which these fishes have lived. The oxygen isotope record produced here, for example, confirms that Paddlefish X-2724 experienced relatively stable environmental conditions during life, implying that the taxon did not migrate between fresh and marine water masses (i.e. (remnants of) the Western Interior Seaway) in the recorded period. Thus, although the utility of the oxygen isotopes of the structurally-bound carbonate in fish bones as a temperature proxy has been far from established, certain very general inferences can be motivated. This aspect has now been addressed under the Stable Isotope Analysis heading of the Methods section.

6. Finally, I question the authors conclusion of “spring” as opposed to summer or even fall – The authors show that there is a clear seasonal cycle in carbon isotopes in figure 2a, however the length of that seasonal cycle is unclear, and it appears that the edge of the bone (interpreted as time of death) is in the middle of the “growth” season, not at the very beginning or very end. Since they present the cyclical data only from a single specimen, and the cycles are varied in magnitude and length, it is not sufficiently clear what this would be showing if additional specimens were included. While the authors say that this is

“spring”, I think that it could be anywhere in the “spring to fall” region based on the data that they present, although likely was closer to late spring or early summer – well in line with Wolfe (1991)’s estimate of “June”.

Response: As Referee #3 indicates, annual growth often occurs from spring to autumn (Castanet et al., 1993), which coincides with the registration of a growth zone in the bone (lines 118-120). However, as now better explained in lines 120-130 osteocyte lacunar density and sizes fluctuates within this growth zone as a function of growth rate, which itself increases in spring, peaks in summer, and decreases in autumn to arrest in winter. We report that, in the months prior to death, osteocyte lacunar density and sizes increase after a pause in growth (i.e. after winter) but had not yet reached the osteocyte lacunar density and sizes associated with the climax of the growth season (which would be summer). This implies that, in all six fishes sampled, growth was truncated during spring.

The carbon isotope record of Paddlefish X-2724 (original Figure 2, now Figure 3) recapitulates this correlated pattern. Although *average* $\delta^{13}\text{C}$ values show a gradual upward trend during the last five full years recorded, the superimposed seasonal fluctuation is clearly recognisable in the relative values. These reveal that peak summer values of $\delta^{13}\text{C}$ consistently increase across all favourable seasons recorded. Nevertheless, the periosteal (i.e. last-formed) bone of X-2724 records a $\delta^{13}\text{C}$ value (-2.46) higher than that recorded in the LAG (winter; -2.84) but significantly lower than the highest value recorded during the preceding summer (-1.32). This archive not only quantifies the temporal position of the moment of death between winter (when the fish was hardly growing) and summer (when $\delta^{13}\text{C}$ would have been substantially higher), but also demonstrates that the carbon isotope record was sufficiently well sampled to yield the required seasonal resolution.

We thank Referee #3 for requesting this clarification, as it allowed us to improve the associated explanation towards the reader.

Some line by line comments:

Line 17 – need a citation here

✓ Smit and Hertogen 1980.

Line 21 – “... leaving the season of the impact unconstrained” – except for paleobotanical evidence, which should be mentioned here

Response: As already explained above, the work by Wolfe (1991) has been disputed, therefore our work is the first to unambiguously resolve the season of the KPg impact event. We now cite several studies that question the materials, methodologies, stratigraphic interpretations and dual-impact hypothesis of Wolfe (1991) in lines 154-159. We believe that the comparable conclusions of that work and ours are purely coincidental (at a 25% chance), we therefore feel that the study should not be mentioned in our abstract.

Line 25 – “These fishes ultimately perished in boreal spring” is repetitive

✓ The repetition has been removed.

Lines 28/29 – There is not evidence in the present manuscript that the boreal spring impact “significantly influenced selective biotic survival across the KPg boundary”. At best, it is a hypothesis that is suggested by the authors in the discussion, and should be couched as such in the abstract.

Response: We agree with Referee #3 that our study cannot conclusively demonstrate how the season of impact explains all selectivity of the extinction but we postulate a framework that could influence selective survival across the KPg boundary. The statement has been amended accordingly. The outcomes of this study allow further testing on such hypotheses, and we believe it will be used and cited in this context.

Paragraph starting on line 31: This paragraph feels somewhat out of place and could be mostly cut, particularly given the context already stated in the abstract. The first sentence about the debate of causal mechanisms is completely unnecessary.

Response: We agree with Referee #3 that this paragraph contained information already presented in the abstract. The first sentence has therefore been removed and the paragraph has been adjusted to avoid repetition.

Line 38 – the citations here are backwards (should be “2, 7” not “7, 2”)

✓ Corrected

Line 52/53 – this sentence is unnecessary, and R. DePalma is mentioned in the acknowledgements. Pers comm not appropriate to cite in text

✓ Corrected

Lines 54-63 – are there any localities beyond Hell Creek that can provide a more nuanced and localized view of seasonality for Tanis? Seasonality can vary greatly a long a latitudinal band depending on proximity to the coast, atmospheric patterns, etc., and

the references used here to “reconstruct” seasonal temperature estimates are not precise temporally or geographically.

Response: We agree with Referee #3 and Referee #4 that more local and more time-constrained studies on seasonality would be much welcomed. However, more geographical- or time-constrained studies on local seasonality during Latest Cretaceous times are presently not available. This lacuna does not infringe on the conclusions of our study but does suggest that our work will be of great value to future studies into uppermost Cretaceous climate records of the Hell Creek Formation.

Lines 87-90 – it is possible to have 3D preservation and also have significant replacement of minerals in the skeletal elements. The fact that there is 3D preservation does not provide evidence for lack of chemical changes.

Response: This is correct, which is why we explicitly tested for potential elemental replacement by means of μ -XRF and confirmed that chemical alteration was negligible. This rationale was, and remains, detailed in the manuscript *before* the 3D presentation is mentioned at all through:

“The degree of preservation of sampled acipenseriform bones was assessed using micro X-ray fluorescence (μ -XRF, see Methods; Supplementary information 2), which is capable of revealing potential taphonomic elemental exchange that may have affected the primary stable isotope composition. The μ XRF maps show that Fe- and Mn-oxides are present in the bone vascular canals and surrounding sediments, but have not invaded the bone apatite.

Detrital sediments, characterised by high concentrations of K and Si, remain restricted to the sediment matrix. The bone apatite conserves a highly homogeneous distribution of P and Ca, which corroborates the unaltered preservation of these apatitic tissues. Skeletal remains of these paddlefishes and sturgeons thus experienced negligible diagenetic alteration, likely due to their rapid burial and possibly aided by early Mn and Fe oxide seam formation.”

Thus, we do not report the exquisite 3D preservation to preclude chemical alteration but rather to motivate that 1) the identity of the recovered elements could be reliably corroborated through association with (surrounding) diagnostic elements and 2) the observed spherule distribution is not the product of taphonomic reworking.

*Lines 93/94 *and* Lines 113-115 – there are often multiple peaks of phytoplankton and zooplankton blooms in temperate systems, spring and fall. Please discuss how you differentiated these, if at all, and why you interpret “spring” as the season rather than summer or fall.*

Response: Although the $\delta^{13}\text{C}$ record presented in original Figure 2a (now Figure 3a) may have captured the effects of both a spring and an autumn bloom in the growth season that occurred circa two years before death, we cared to refrain from over-

interpretation.

Nevertheless, the cumulative effect of potential discrete blooms that Referee #3 suggests may have occurred during the growing seasons evidently translated to a well-defined annual (rather than sub-annual) $\delta^{13}\text{C}$ cyclicity across the final years of the Cretaceous that abruptly stops at a value most consistent with spring. Moreover, the independent osteohistological records of all six fishes produce exactly the same signal: growth has commenced after winter but bone corresponding to the climax of the growth season (largest osteocyte lacunar density; largest osteocyte lacunae) had not yet been deposited (see also our response to Remark #7). In both the $\delta^{13}\text{C}$ and the growth records, recording appears to have stopped relatively soon after the preceding unfavourable period (i.e. winter). This itself is not conclusive evidence for “spring”, as discussed earlier, but is fully consistent with the recurring osteohistological zonation in growth bands, the annual cyclicity in osteocyte lacunar properties, and the annual pattern in $\delta^{13}\text{C}$ values across the preceding years that all place the moment of death in spring.

Line 122 – these references do not seem to support this sentence

Corrected

Final paragraph – this is a lot of interesting speculation, and is missing citations to relevant literature that has discussed these geographic and ecosystem selectivity regimes.

Response: Although we aimed to appropriately discuss relevant influences of a boreal spring impact on selective survival of the KPg extinction in the final paragraph, we agree with Referee #3 that this section was not yet fully referenced and may therefore have appeared somewhat speculative. We have therefore modified the referred discussion of the implications into focused sections that address direct (terrestrial) effects of the impact relevant on a seasonal timescale (lines 146-152), seasonal vulnerability of terrestrial taxa (lines 153-159), seasonal dormancy and/as a potential explanation for survival during austral autumn (lines 160-168), and the projected value of our study towards future efforts in resolving different extinction mechanisms acting on short- and long-term timescales (lines 169-176), including the Anthropocene extinction, as well as geographical regimes (i.e. northern vs southern hemisphere). The discussion now offers a fully referenced context for the implications of and opportunities offered by our study.

Line 307 – please include specific permanent URLs to the specimens

Response: All tomographic data used in the study will be made openly available upon publication through paleo.esrf.fr. This is the established ESRF heritage database for palaeontology, evolutionary biology, and archaeology that facilitates intuitive access to

synchrotron tomographic scanning data associated with published reports. The data are presently not yet publicly archived in order to secure the novelty of our study but could be, upon request, confidentially shared with the referees and/or editors prior to publication for corroboration.

Figure 1 – this did not translate well to black and white/printing, although the googly eye placement made me smile. I appreciate the necessity of this figure, and panel B is particularly interesting and well done.

Response: We are pleased to learn that Referee #3 appreciates Figure 1.

Figure 2 – please see comments above. This figure is a good start, but is missing key information.

- ✓ Original Figure 2 (now Figure 3) has been improved following the suggestions of Referee #3 and #4.

Comments on supplementary information: many of the scale bars are extremely hard to read because the text is too small (e.g. pages 3, 4, 6, 7, 9, 10, 12, 13, 15, 16, 18, 19) Page 39 (SI) has some extra text on the image

- ✓ The Supplementary Information has been improved following the suggestions of Referee #3.

References

DePalma, Robert A., et al. "A seismically induced onshore surge deposit at the KPg boundary, North Dakota." *Proceedings of the National Academy of Sciences* 116.17 (2019): 8190-8199.

Pucéat, E., Joachimski, M.M., Bouilloux, A., Monna, F., Bonin, A., Motreuil, S., Morinière, P., Hénard, S., Mourin, J., Dera, G. and Quesne (2010). Revised phosphate–water fractionation equation reassessing paleotemperatures derived from biogenic apatite. *Earth and Planetary Science Letters*, 298(1-2), 135-142.

Kolodny, Yehoshua, Boaz Luz, and Oded Navon. "Oxygen isotope variations in phosphate of biogenic apatites, I. Fish bone apatite—rechecking the rules of the game." *Earth and Planetary Science Letters* 64.3 (1983): 398-404.

Pellegrini, M., Lee-Thorp, J. A., & Donahue, R. E. (2011). Exploring the variation of the $\delta^{18}\text{O}_{\text{p}}$ and $\delta^{18}\text{O}_{\text{c}}$ relationship in enamel increments. *Palaeogeography, Palaeoclimatology, Palaeoecology*, 310(1-2), 71-83.

Vennemann, T. W., et al. "Isotopic composition of recent shark teeth as a proxy for

environmental conditions." *Geochimica et Cosmochimica Acta* 65.10 (2001): 1583-1599.

Wolfe, Jack A. "Palaeobotanical evidence for a June 'impact winter' at the Cretaceous/Tertiary boundary." *Nature* 352.6334 (1991): 420-423.

Nichols, Douglas J. "Plants at the K/T boundary." *Nature* 356.6367 (1992): 295-295.

Hickey, Leo J., and McWeeney, Lucinda J. Response to Wolfe, J.A. 1991. Palaeobotanical evidence for a June "impact winter" at the Cretaceous/Tertiary boundary. *Nature*, 356: 295–296.

McIver, Elisabeth. E. "Paleobotanical evidence for ecosystem disruption at the Cretaceous- Tertiary boundary from Wood Mountain, Saskatchewan, Canada." *Canadian Journal of Earth Sciences* 36.5 (1999): 775-789.

Upchurch, Garland R., Barry H. Lomax, and David J. Beerling. "Paleobotanical evidence for climatic change across the Cretaceous-Tertiary boundary, North America: Twenty years after Wolfe and Upchurch." *Courier-Forschungsinstitut Senckenberg* 258 (2007): 57.

Castanet, J. "Bone and individual aging." *Bone growth* (1993): 245-283.

Referee #4 (Remarks to the Author):

Dear editor,

The manuscript by During et al. presents a unique bone histological record from several fossil fishes from the spectacular Tanis Cretaceous-Paleogene boundary seiche deposit. Based on impact spherules lodged in their gills, it can be assumed that these fish died on the day of the Chicxulub asteroid impact. The bone histology and stable carbon isotope record of one of these fishes suggests that the death of the studied animal occurred during boreal spring time, providing a tentative constrain on the season in which the Chicxulub impact occurred.

The study is highly original and absolutely significant, both for the Cretaceous-Paleogene boundary impact community as well as for a wider audience. The approach is valid, data of good quality, with appropriate treatment of uncertainties, presentation is simple but clear. Perhaps the main figure could use some additional labels, to guide the less-informed reader a bit better (see attached list of comments and suggestions).

The presented data support the main conclusion of the manuscript, that is that the Chicxulub impact occurred in boreal spring. Nevertheless, with only one stable isotope record presented, the robustness remains difficult to assess. According to the Materials & Methods section, two specimens were measured for isotopes, while only one is provided in the manuscript. If both records, both sampled up to the outermost growth increment, would be presented, this would considerably strengthen the case. While here and there the choice of references might not be optimal (see attached list of comments and suggestions), in general the authors appropriately credit previous work.

All in all, the manuscript is clearly written, the abstract clear and appropriate and the introduction and conclusions are all easy to follow. Even though there are several minor issues with the (re)presentation of the data, the high quality of the research, its broad significance and major potential impact lead me to recommend this manuscript to be accepted with revisions.

List:

Line 18: *Lowery et al. (2018) is a study on the biological recovery after the impact, not a really good reference on the extinction rates. For this, one better refers to the studies of Jablonski, Alroy, Bambach, Raup and Sepkoski.*

✓ Corrected

Lines 26-27: *I don't know if the term "diversity" is in place here. Maybe "among latest Cretaceous biotic clades" would better catch the meaning of this sentence?*

Corrected

Lines 40-41: *The Vellekoop et al. (2019) paper is not about the aftermath of the KPg boundary impact. For these the Vellekoop et al. 2014 (PNAS) 2016 (Geology) and 2018 (Geology) papers would be better references.*

✓ Corrected

Lines 48-49: *benthic foraminifera have not been recorded in the DePalma paper? Wheredoes this new evidence come from?*

Response: We now realise that benthic foraminifera were indeed not yet mentioned in DePalma et al. (2019), although they are abundantly present in the Tanis assemblage. Several representative examples are now depicted in Supplementary Information 1.

Lines 58-60: *are the rather ancient studies by Hallam (1985) and Golovneva (2000) reallythe most recent estimations of the regional climate?*

Response: To our knowledge, no appropriate and more recent studies on the regional climate are presently available. Virtually all climate reconstructions focus on Mean Annual Temperatures rather than sub-annual (such as seasonal) climate conditions.

Line 60: *a Stage is a chronostratigraphic unit, the word "latest" indicates that the authors are discussing time (=geochronology) here. Therefore, this should be "the latest age of theCretaceous ("*

✓ Corrected

Line 63: *the Hell Creek Formation is a formation, not a site. Technically, the Tanis site alsocomprises the Hell Creek...*

✓ Corrected

Lines 258-260: *Why is only the isotope data of X-2724 plotted? Please also provide the record of MDX-3, either in Figure 2, or in the supplements. If both records, both sampled*

upto the outermost growth increment, show the same pattern, this would highly strengthen the case.

Response: As also discussed in our response to Remarks #2 by Referee #3, attempts to recover representative incremental samples were made on all six fish elements studied here. However, only MDX-3 and X-2724 retrieved sufficient apatite powder for stable isotope analyses, as the other elements preserved highly convoluted interior growth patterns (sturgeon fin spines X-2743M and X-2744M) or very narrowly spaced growth marks (paddlefish dentaries X-2733a and X-2733b). These conditions prevented us from retrieving samples that confidently represent the discrete sub-annual growth stages required for reliable recovery of the seasonal growth patterns. MDX-3 did yield incremental samples, but these were digested during analysis without producing results, as sample weights proved too low to produce a sufficient signal amplitude for stable isotopic analyses. This issue was circumvented for X-2724, which did produce a reliable isotope record along its growth history.

As also demonstrated in our responses to Remarks #3 and #7 by Referee #3, the osteohistological life history records of all six individual fishes studied here conclusively converge on a spring death. The isotopic record of X-2724 independently corroborates this conclusion and is not presented in support of additional claims founded exclusively on the isotope analysis that may be argued to require validation of a second isotopic record. As the bone histology of six fishes (that are known to have perished simultaneously; DePalma et al., 2019) points to a spring death and the isotopic record of one of those fishes also points to a spring death, we believe to already provide two independent and mutually reinforcing arguments for a spring death.

Figure 2: *For the less-informed reader, please indicate the interpreted seasons in the figure, not just the LAGs.*

✓ Corrected (N.B. original Figure 2 is now Figure 3)

Reviewer Reports on the First Revision:

Referee #1 (Remarks to the Author):

I reviewed a previous version of this manuscript, and did push forward some concerns related to my fields of expertise (namely, systematics, comparative anatomy and bone histology of fossil and extant ray-finned fishes).

While some of these concerns were important, I did not think they warranted a rejection of the manuscript. I am therefore happy to see it has been successfully resubmitted.

Since the general structure of the MS in the previous version was sound, with very few minor writing issues that have since been corrected, I will elaborate on the changes that have been made in response to my previous comments.

1) Justification for combining physical thin sections and virtual synchrotron histology:

I feel that the additional text, along with the few more references that have been added, is now adequately justifying this methodological approach.

2) Clarifications on annual growth:

I feel these are now much clearer and help understanding the text. I think Figure 3 is now clearer, but would have liked to see changes in osteocyte lacuna volumes highlighted more clearly (see below).

3) Identification of the specimens as acipenseriforms: The new figures in Supplements are very good at lowering any ambiguity, especially Fig. S2.2.

4) Clarity of Fig. 1: It is now clearer thanks to the added legends. The other view presented in the supplements is useful too.

5) Identity of the spines: Fig. S2.2 supports a sturgeon identity more clearly. I agree with the authors that the seasonal signal is supported by all specimens, therefore this identity is not extremely important.

Fig. 3: Changes in osteocyte lacuna density are now more visible thanks to the green shading. However, changes in size are not that clear. A better way to visualize it would have been to segment out the lacunae, using a color gradient proportional to volume. I understand that it is time-consuming but this variation is abundantly discussed in the text and a nice visualization of lacunae would go a long way to make the results more impactful. I have to note that this size variation is somewhat more clearly visible in the Supplements (Fig. S5.2), so at the very least pointing to this figure in the main text would be warranted.

All in all, I am satisfied with the changes that have been made to the manuscript, as far as my fields of interest are concerned. My main wish now would be a clearer visualization of osteocyte lacunae, if time allows it.

Thank you very much for inviting me to review this interesting study!

Donald Davesne, Museum für Naturkunde, Berlin, Germany

Referee #3 (Remarks to the Author):

During et al present, in their manuscript "The Mesozoic terminated in boreal spring", compelling and novel evidence for the seasonality of the Chicxulub impact, using a combination of bone histology and stable isotopes on fossil fish which they interpret to have died in the immediate (minutes to hours) aftermath of the impact event. The method, to my knowledge, is novel, and the manuscript they present, significantly improved from the prior version which I reviewed previously, presents a clear set of arguments supporting their conclusions. It is an innovative series of experiments on an exceptional fossil assemblage, and I think that the authors have come a long way in their manuscript and presentation of these data. I particularly appreciate the improvements to Figures 2/3.

The biggest thing that I think is (still) missing is a quantification of the density/color banding clearly showing the cycles that underpin the manuscript – the additional labeled images are helpful, but they do not provide sufficient data. This would ideally take the form of a graph that shows a transect along the bone on the x-axis and intensity or color on the Y-axis. These graphs are straightforward to make using ImageJ or other free image processing software, and can be overlain on the images along the transect that produced the graph or appended as separate panels. I think Figure 3 in particular would benefit from having a density/intensity panel beyond just the images themselves, and that all the panels in Figure 2 as well as the supplemental figures, could benefit from having intensity graphs overlaying them.

The red arrows that have been added to some figures are somewhat helpful, though I have an extremely difficult time seeing what bands they are specifically pointing to (and I have spent several years of my life looking at x-rays and bands of various types of fossils, so I am very familiar with this type of image). The red arrows are also not present on all of the images, and at least some of the red arrows appear to be pointing to what look like cracks in the fossil, rather than specific growth bands. Quantification of the growth lines would help this considerably.

Additionally, the authors spend a lot of space discussing the geographical selectivity of terrestrial extinctions, but they might also consider this paper looking at geographic patterns of coccolithophore (nannofossil) survival at the K/Pg that show a clear hemispheric signal – this could strengthen their discussion by expanding their scope to the marine realm (there are many others discussing marine extinction geographic selectivity, see the work of Hull et al 2011, but this one aligns well with their hypotheses of hemispheric selectivity):

S. J. Jiang, T. J. Bralower, M. E. Patzkowsky, L. R. Kump, J. D. Schueth, Geographic controls on nannoplankton extinction across the Cretaceous/Palaeogene boundary. *Nat Geosci* 3, 280-285 (2010).

Line-by-line notes:

Line 29: “most selective” implies a specific ecologic or taxonomic selectivity. I think here the authors mean “most extreme” or “largest” (high percentage of total extinction), rather than “most selective, ecologically. Please provide a citation for taxonomic or ecological selectivity, or rephrase to be more clear about the use/meaning of “selective”.

Line 49: “climatic turnover” is unclear.

Line 121 – ‘the six fishes’ – X-2733a and X-2733b are the same individual are they not? That would make 5 fishes

Line 151 – remove “conclusive” – it is supportive of a spring death, but “the conclusive” seems a bit over-stepped.

Figure 1b – please add a label to some of the spherules, as you have labeled the gill rakers and gill arches; Please also add on the figure what the scale bar’s length is.

Figure 2 – please add density/intensity plots as mentioned above. Caption for Figure 2 also needs to specify what the red arrows are.

Figure 3 – This is a great addition to the manuscript, but needs a panel with density/intensity cycles that are the primary data.

Referee #4 (Remarks to the Author):

Dear editor,

The revised manuscript by Doring et al. presents a unique bone histological record from several fossil fishes from the spectacular Tanis Cretaceous-Paleogene boundary seiche deposit. Based on impact spherules lodged in their gills, it can be assumed that these fish died on the day of the Chicxulub asteroid impact. The bone histology and stable carbon isotope record of one of these fishes suggests that the death of the studied animal occurred during boreal spring time, providing a tentative constrain on the season in which the Chicxulub impact occurred.

The study is highly original and absolutely significant, both for the Cretaceous-Paleogene boundary impact community as well as for a wider audience. The approach is valid, data of good quality, with appropriate treatment of uncertainties, presentation is simple but clear. Perhaps the main figure could use some additional labels, to guide the less-informed reader a bit better (see suggestions below).

The presented data support the main conclusion of the manuscript, that is that the Chicxulub impact occurred in boreal spring. In the revised version of the manuscript, the focus is more on the histology and less on the stable isotopes, as there apparently is only one stable isotope record available. This does mean that the robustness of the isotopic signal remains difficult to assess. Nevertheless, as the histology is now used as the main line of evidence, this should be sufficient.

All in all, the manuscript is clearly written, the abstract clear and appropriate and the introduction and conclusions are all easy to follow. The authors have tackled most of my previous issues with the manuscript, leaving only a few minor issues. The broad significance of the research and major potential impact lead me to recommend this manuscript to be accepted with minor revisions.

Comments and suggestions:

Lines 139-140: Maybe also explain why isotopic data from sturgeon pectoral fin spines could not be secured?

Lines 376-377: Is the term "isotopic determinations" really the right choice of words here?

Figure 2: please indicate in the caption or in the figure what the red arrows indicate.

Figure 3: as suggested in my previous review, please indicate the interpreted seasons (winter-spring-summer-fall) in this figure. Not just a "colour gradient reflects the theoretical range", but indicate which growth lines are formed during which seasons.

Author Rebuttals to First Revision:

Referee #1 (Remarks to the Author):

I reviewed a previous version of this manuscript, and did push forward some concerns related to my fields of expertise (namely, systematics, comparative anatomy and bonehistology of fossil and extant ray-finned fishes).

While some of these concerns were important, I did not think they warranted a rejection of the manuscript. I am therefore happy to see it has been successfully resubmitted.

Since the general structure of the MS in the previous version was sound, with very few minor writing issues that have since been corrected, I will elaborate on the changes that have been made in response to my previous comments.

1) Justification for combining physical thin sections and virtual synchrotron histology: I feel that the additional text, along with the few more references that have been added, is now adequately justifying this methodological approach.

2) Clarifications on annual growth:

I feel these are now much clearer and help understanding the text. I think Figure 3 is now clearer, but would have liked to see changes in osteocyte lacuna volumes highlighted more clearly (see below).

3) *Identification of the specimens as acipenseriforms: The new figures in Supplements are very good at lowering any ambiguity, especially Fig. S2.2.*

4) *Clarity of Fig. 1: It is now clearer thanks to the added legends. The other view presented in the supplements is useful too.*

5) *Identity of the spines: Fig. S2.2 supports a sturgeon identity more clearly. I agree with the authors that the seasonal signal is supported by all specimens, therefore this identity is not extremely important.*

Fig. 3: Changes in osteocyte lacuna density are now more visible thanks to the green shading. However, changes in size are not that clear. A better way to visualize it would have been to segment out the lacunae, using a color gradient proportional to volume. I understand that it is time-consuming but this variation is abundantly discussed in the text and a nice visualization of lacunae would go a long way to make the results more impactful. I have to note that this size variation is somewhat more clearly visible in the Supplements (Fig. S5.2), so at the very least pointing to this figure in the main text would be warranted. All in all, I am satisfied with the changes that have been made to the manuscript, as far as my fields of interest are concerned. My main wish now would be a clearer visualization of osteocyte lacunae, if time allows it.

Response. We thank Referee #1 for these comments and have indeed included a more explicit reference in the main text to the better visible larger osteocyte lacunae in Supplementary Figure S5.2. We also agree that segmentation of osteocyte lacunar volumes offers a valuable avenue towards resolving cyclical growth, as has been demonstrated in his own pioneering work. Following the invitation by Referee #1, we have therefore applied the protocol outlined in Devasne et al. (2020) on the scanning data representing the dentary of paddlefish X-2724. Since the resolution available to us (voxel size of 4.35 μm ; appropriate for assessing growth marks and osteocyte lacunar distributions) is sixfold lower than that used by Devasne et al. (2020; voxel size of 0.7 μm), our result should be considered with appropriate care. Closely-spaced (large) osteocyte lacunae may now be conjoined and additional phenomena in the broad size range of osteocyte lacunae may be incidentally included in the visualised distribution. Moreover, the osteocyte lacunar borders follow artifactual gradients (rather than a discrete line) that scales with voxel size. Because the outermost feature fringe contributes disproportionately to recovered volumes, these values are skewed relative to the original osteocyte lacunar volumes, which likely produces exaggerated volume values. Therefore, although all rendered features were extracted with a single thresholding operation, volume values are best appreciated in a comparative context. Nevertheless, relative patterns are conservatively retained.

The distribution of relative osteocyte lacunar volumes is fully consistent with the observations made on traditional osteohistological thin sections, virtual thick sections, and the stable carbon isotope curve. Cyclical fluctuations in osteocyte lacunar volume describe seven growth zones (abundant large osteocyte lacunae; spring and summer), all followed by well-defined annuli and LAGs (less densely-packed, smaller osteocyte lacunae). Towards the periosteal surface, the onset of the ultimate growth season is expressed as a thin layer of enlarged osteocyte lacunae. This agrees with the interpretation of death in the early stage of the favourable growth season (i.e. spring).

[Figure redacted]

Thank you very much for inviting me to review this interesting study! Donald Davesne, Museum für Naturkunde, Berlin, Germany

We thank Dr. Donald Davesne for providing insightful comments and valuable discussion on our manuscript.

Referee #3 (Remarks to the Author):

During et al present, in their manuscript “The Mesozoic terminated in boreal spring”, compelling and novel evidence for the seasonality of the Chicxulub impact, using a combination of bone histology and stable isotopes on fossil fish which they interpret to have died in the immediate (minutes to hours) aftermath of the impact event. The method, to my knowledge, is novel, and the manuscript they present, significantly improved from the prior version which I reviewed previously, presents a clear set of arguments supporting their conclusions. It is an innovative series of experiments on an exceptional fossil assemblage, and I think that the authors have come a long way in their manuscript and presentation of these data. I particularly appreciate the improvements to Figures 2/3.

The biggest thing that I think is (still) missing is a quantification of the density/color banding clearly showing the cycles that underpin the manuscript – the additional labeled images are helpful, but they do not provide sufficient data. This would ideally take the form of a graph that shows a transect along the bone on the x-axis and intensity or color on the Y-axis.

These graphs are straightforward to make using ImageJ or other free image processing software, and can be overlain on the images along the transect that produced the graph or appended as separate panels. I think Figure 3 in particular would benefit from having a density/intensity panel beyond just the images themselves, and that all the panels in Figure 2 as well as the supplemental figures, could benefit from having intensity graphs overlaying them.

Response: We have created a bone cellular density map following the protocol of Sanchez et al. (2013). This density map presents bone regions with high cellular density in orange and regions with low cellular density in purple, and now replaces Figure 3c and has been included in full size as Supplementary Information 5.6. The new figure strongly supports the assumption of higher cellular densities in growth zones (i.e. bone deposited during favourable seasons, namely spring and summer).

A graph plotting colour intensity along an endosteal-periosteal bone transect visualised using light microscopy has proven unsuitable for resolving fluctuations in bone colour resulting from growth. This method is incapable of distinguishing between the bone matrix and included phenomena, such as vascular canals, osteocyte lacunae, Sharpey's fibers, and microfractures. Eliminating such features through an appropriately-tuned image filter results in profound blurring that obscures the growth signal itself. For example, abundant (dark) osteocyte lacunae in the growth zones were found to largely compensate for the inherently lighter bone matrix in the zones, rendering them poorly resolved against the

darker matrix of the adjacent annuli.

The red arrows that have been added to some figures are somewhat helpful, though I have an extremely difficult time seeing what bands they are specifically pointing to (and I have spent several years of my life looking at x-rays and bands of various types of fossils, so I am very familiar with this type of image). The red arrows are also not present on all of the images, and at least some of the red arrows appear to be pointing to what look like cracks in the fossil, rather than specific growth bands. Quantification of the growth lines would help this considerably.

Response: The red arrows consistently indicate LAGs, which occasionally coincide with secondary fractures that are known to preferentially propagate along LAG surfaces due to the local lamellar organisation of the bone matrix. This phenomenon is often observed in fossil and even (sub-) recent bone.

Following the preferential growth seasons, skeletal growth slows down. During slow growth, collagen fibre orientations alternate between circumferential and longitudinal directions, forming a 'plywood' bone matrix. This "lamellar bone" tissue typically forms the annuli in zonal bone that is deposited during autumn and early winter. Following the annulus, growth ceases and a LAG is formed.

In spring, bone deposition recommences through the formation of a quickly-deposited bone "zone", the density and structure of which differs from the slowly-deposited bone of the preceding annulus. The resulting profound disparity in bone structure and density creates a poorly amalgamated interface that preferentially hosts crack propagation. As a consequence, some of the LAGs in our samples are emphasised by post-depositional crack formation. The red arrows indicate these LAGs, which is now explicitly detailed in the caption of Figure 2 as well.

Additionally, the authors spend a lot of space discussing the geographical selectivity of terrestrial extinctions, but they might also consider this paper looking at geographic patterns of coccolithophore (nannofossil) survival at the K/Pg that show a clear hemispheric signal – this could strengthen their discussion by expanding their scope to the marine realm (there are many others discussing marine extinction geographic selectivity, see the work of Hull et al 2011, but this one aligns well with their hypotheses of hemispheric selectivity):

S. J. Jiang, T. J. Bralower, M. E. Patzkowsky, L. R. Kump, J. D. Schueth, Geographic controls on nannoplankton extinction across the Cretaceous/Palaeogene boundary. NatGeosci 3, 280-285 (2010).

Response: We thank Referee #3 for suggesting this highly appropriate work that emphasises the relevance of our study. Our discussion presently focuses on the terrestrial rather than the marine realm. Given the limitations on article length and the amount of references suggested, addressing this work would imply relinquishing at least one of the studies currently presented in our discussion. Nevertheless, if the editor permits us, we will gratefully include this perspective in our manuscript.

Line-by-line notes:

Line 29: "most selective" implies a specific ecologic or taxonomic selectivity. I think here the authors mean "most extreme" or "largest" (high percentage of total extinction), rather than "most selective, ecologically. Please provide a citation for taxonomic or ecological

selectivity, or rephrase to be more clear about the use/meaning of “selective”.

Response: We understand that this wording may be confusing. Therefore, we have adjusted the sentence to recall to the original wording from the source reference (Raup, 1986).

Line 49: “climatic turnover” is unclear.

✓ Corrected.

Line 121 – ‘the six fishes’ – X-2733a and X-2733b are the same individual are they not? That would make 5 fishes

Response: We understand this assumption by Referee #3. However, the two referred dentaries actually represent two individual paddlefishes. Specimens X-2733a and X-2733b were recovered from the field in the same matrix block (X-2733). Preparation revealed the presence of not one, but two fishes (i.e. X-2733a and X-2733b) that were preserved on top of each other. This has now been clarified in line 347 in the Methods section.

Line 151 – remove “conclusive” – it is supportive of a spring death, but “the conclusive” seems a bit over-stepped.

✓ Corrected, the carbon isotope data indeed corroborated the spring death.

Figure 1b – please add a label to some of the spherules, as you have labeled the gill rakers and gill arches; Please also add on the figure what the scale bar’s length is.

Response: We have implemented the helpful suggestion by Referee #3.

Figure 2 – please add density/intensity plots as mentioned above. Caption for Figure 2 also needs to specify what the red arrows are.

Response: We have now explained the red arrows in Figure 2, which we had overlooked earlier – thank you for drawing our attention to this. A density map has now been added to Figure 3c and as Supplementary information 5.6.

Figure 3 – This is a great addition to the manuscript, but needs a panel with density/intensity cycles that are the primary data.

Response: We thank Referee 3 for this compliment, we have followed the suggestion and added a density map in Figure 3C.

Referee #4 (Remarks to the Author):

Dear editor,

The revised manuscript by During et al. presents a unique bone histological record from several fossil fishes from the spectacular Tanis Cretaceous-Paleogene boundary seiche deposit. Based on impact spherules lodged in their gills, it can be assumed that these fish died on the day of the Chicxulub asteroid impact. The bone histology and stable carbon isotope record of one of these fishes suggests that the death of the studied animal occurred during boreal spring time, providing a tentative constrain on the season in which the Chicxulub impact occurred.

The study is highly original and absolutely significant, both for the Cretaceous-Paleogene boundary impact community as well as for a wider audience. The approach is valid, data of good quality, with appropriate treatment of uncertainties, presentation is simple but clear. Perhaps the main figure could use some additional labels, to guide the less-informed reader a bit better (see suggestions below).

The presented data support the main conclusion of the manuscript, that is that the Chicxulub impact occurred in boreal spring. In the revised version of the manuscript, the focus is more on the histology and less on the stable isotopes, as there apparently is only one stable isotope record available. This does mean that the robustness of the isotopic signal remains difficult to assess. Nevertheless, as the histology is now used as the main line of evidence, this should be sufficient.

All in all, the manuscript is clearly written, the abstract clear and appropriate and the introduction and conclusions are all easy to follow. The authors have tackled most of my previous issues with the manuscript, leaving only a few minor issues. The broad significance of the research and major potential impact lead me to recommend this manuscript to be accepted with minor revisions.

Comments and suggestions:

Lines 139-140: Maybe also explain why isotopic data from sturgeon pectoral fin spines could not be secured?

Response: We thank Referee 4 for the comments and suggestions. The main text now includes a reference to the methods section regarding the Micromill where the issue of sample amounts has been addressed in detail.

Lines 376-377: Is the term "isotopic determinations" really the right choice of words here?

Response: We believe to have used the correct terminology.

Figure 2: please indicate in the caption or in the figure what the red arrows indicate.

✓ Corrected.

Figure 3: as suggested in my previous review, please indicate the interpreted seasons (winter-spring-summer-fall) in this figure. Not just a "colour gradient reflects the theoretical range", but indicate which growth lines are formed during which seasons.

Response: Ideally we would indeed include all recorded/interpreted seasons in Figure 3. However, we believe this would obscure the image as there is no space for more text. To circumvent this, we have updated Figure 3C with the osteocyte lacunar density map that captures highest densities in summer (orange) and lowest densities in winter (purple). Colour gradients are preferred as they reflect the true nature of the specimens and not exclusively our interpretation.

Reviewer Reports on the Second Revision:

Referee #1 (Remarks to the Author):

I reviewed this manuscript for the third time, and am happy to note that my remarks and concerns (as well as those of the other reviewers) have been adequately addressed.

Fig. 3 now does a good job at clearly showing variation in osteocyte lacuna density, which was a concern that was raised before.

This figure reads very well right now.

I appreciate you taking time for producing the visualization of cell volumes, as I suggested. This figure conveys the information quite well, however it is not in the manuscript nor in the supplements at the moment. I realise you didn't include it due to uncertainties in the measurements, because resolution is not sufficient to properly image the lacunae. These concerns are perfectly valid; however I feel that it may still be beneficial to include it in the supplements, as long as a proper warning is clearly given. You can provide the same explanation that you gave in the rebuttal letter, with a few tweaks. Ultimately, it is your call and if you don't feel comfortable doing so, it is your decision.

Congratulations for this exciting research, and thank you for taking my reviews into account so thoroughly.

Donald Davesne,
Museum für Naturkunde, Berlin, Germany

Referee #3 (Remarks to the Author):

During et al have continued to make improvements to both the manuscript and the figures. I particularly appreciate the addition of Figure 3c, which nicely demonstrates zones of higher and lower density. Overall I think that this manuscript is nearly ready for publication, and as I have said in previous reviews, is an elegant use of a rare and exceptionally preserved fossil assemblage, alongside novel methodology, to solve an important question in paleontology.

However, I respectfully disagree with the authors' claim that the red arrows consistently and clearly indicate LAGs and are "easily recognized" (line 79) – I have spent considerable time looking at all of these images trying to convince myself of this, and agree with the authors only ~80% of the time – both in the sense that I find what appear to be LAGs that are not marked with red arrows, and the authors mark some that I cannot convince myself of. The data are difficult to discern in some cases, as biological data often are. I do not believe that this is fatal to the manuscript, and indeed, it is necessary to show this uncertainty, as it acknowledges the challenges of working with this material. This is why I have been pushing for colour-intensity graphs and other similar information (e.g. Fig 3c) which capture, quantify, and acknowledge some of that uncertainty. These assertions need to be backed up with something more than red arrows photoshopped onto a photograph.

I made a very quick figure from a screenshot of Figure 2c, demonstrating the utility and appropriate-ness of graphing colour-intensity using ImageJ's "Analyze/Plot Profile" option on a line that was approximately perpendicular to the axis of growth. I note that while smoothing does lose a little bit of the variability in the growth patterns as noted by the authors in their rebuttal letter, a simple moving-average filter enhances the overall patterns of light- and dark- bands, rather than obscuring it. Please see the included figure. I note that LAGs are defined by a sharp change in the value of the colour intensity, rather than a strict threshold value. I think that it is necessary to include a profile plot for each of the specimens imaged and analyzed, along with arrows indicating the interpreted LAGs. This could be done either in the main text or the supplement, and would greatly improve the quality of the manuscript, while demonstrating the utility and fidelity of this novel methodology.

Line-by-line notes:

Line 27 – you use the word "impact" throughout the abstract/manuscript to mean different things (e.g. "impact on the Yucatan Peninsula" vs. "impacted 76% of species") – a different word/phrase here, e.g. "extirpated" or "vanquished" or more simply "resulted in the loss of" would be both more precise and help reduce repetition and confusion.

Line 175 – "than e.g. birds" is a bit confusing. Perhaps rephrase "than other groups (e.g. birds)"

Line 187 – you use birds here again as the only example of surviving lineages. Could expand this a bit to include another group or two (e.g. mammals, fish, etc)

Author Rebuttals to Second Revision:

Referee #1 (Remarks to the Author):

I reviewed this manuscript for the third time, and am happy to note that my remarks and concerns(as well as those of the other reviewers) have been adequately addressed.

Fig. 3 now does a good job at clearly showing variation in osteocyte lacuna density, which was a concern that was raised before.

This figure reads very well right now.

I appreciate you taking time for producing the visualization of cell volumes, as I suggested. This figure conveys the information quite well, however it is not in the manuscript nor in the supplements at the moment. I realise you didn't include it due to uncertainties in the measurements, because resolution is not sufficient to properly image the lacunae. These concerns are perfectly valid; however I feel that it may still be beneficial to include it in the supplements, as long as a proper warning is clearly given. You can provide the same explanation that you gave in the rebuttal letter, with a few tweaks. Ultimately, it is your call and if you don't feel comfortable doing so, it is your decision.

Congratulations for this exciting research, and thank you for taking my reviews into account so thoroughly.

*Donald Davesne,
Museum für Naturkunde, Berlin, Germany*

We thank Donald Davesne for his helpful comments and for encouraging us to include the osteocyte lacunar volume figure in the Supplementary Information.

Referee #3 (Remarks to the Author):

During et al have continued to make improvements to both the manuscript and the figures. I particularly appreciate the addition of Figure 3c, which nicely demonstrates zones of higher and lower density. Overall I think that this manuscript is nearly ready for publication, and as I have said in previous reviews, is an elegant use of a rare and exceptionally preserved fossil assemblage, alongside novel methodology, to solve an important question in paleontology.

However, I respectfully disagree with the authors' claim that the red arrows consistently and clearly indicate LAGs and are "easily recognized" (line 79) – I have spent considerable time looking at all of these images trying to convince myself of this, and agree with the authors only ~80% of the time – both in the sense that I find what appear to be LAGs that are not marked with red arrows, and the authors mark some that I cannot convince myself of. The data are difficult to discern in some cases, as biological data often are. I do not believe that this is fatal to the manuscript, and indeed, it is necessary to show this uncertainty, as it acknowledges the challenges of working with this material. This is why I have been pushing for colour-intensity graphs and other similar information (e.g. Fig 3c) which capture, quantify, and acknowledge some of that uncertainty. These assertions need to be backed up with something more than red arrows photoshopped onto a photograph.

I made a very quick figure from a screenshot of Figure 2c, demonstrating the utility and appropriate-ness of graphing colour-intensity using ImageJ's "Analyze/Plot Profile" option on a line that was approximately perpendicular to the axis of growth. I note that while smoothing does lose a little bit of the variability in the growth patterns as noted by the authors in their rebuttal letter, a simple moving-average filter enhances the overall patterns of light- and dark-bands, rather than obscuring it. Please see the included figure. I note that LAGs are defined by a sharp change in the value of the colour intensity, rather than a strict threshold value. I think that it is necessary to include a profile plot for each of the specimens imaged and analyzed, along with arrows indicating the interpreted LAGs. This could be done either in the main text or

the supplement, and would greatly improve the quality of the manuscript, while demonstrating the utility and fidelity of this novel methodology.

We appreciate that Referee #3 took the time to produce the included colour intensity plots. Growth Marks (GMs) are clearly discernible across the whole sectioned fin spines and jaw bones under a microscope using transmitted light (Supplementary Information 3) and we initially explored a comparable approach using colour intensity (i.e. brightness) to log GMs as pairs of opaque and translucent (i.e. dark and light, respectively) layers. We found that the resulting colour intensity curves are strongly affected by the local and variable presence of secondary (darker) textures associated with vascular canals, osteocyte lacunae, Sharpey's fibres, and microscopic cracks. Therefore, colour intensity transects (measuring a few micrometres in width) do not offer an accurate representation of periodic cyclicity registered in the bone tissue. It should furthermore be noted that, as a consequence of the complex morphologies of the dentaries and pectoral fin spines, the GMs are not uniform in thickness and shape. The placement of an ideal colour intensity transect would need to be perfectly perpendicular to all growth marks but not intersect with any of the referred darker 'inclusions', which is virtually impossible. Because the middle of each growth zone is most densely populated with aforementioned darker features (and, thus, pixels), we found that "averaging" local values can result in two dark layers within a single GM that may be misconstrued to reflect two annulus/LAG complexes while the original slide clearly shows a single GM is represented.

Increasing the width of the transect and/or selective pixel consideration may alleviate some of these problems but the complex bone geometry then blurs the narrow transitions between adjacent components within the GMs.

We respectfully maintain that the established protocols for evaluating osteohistological thin sections offer the best-understood insights into the growth histories recorded in modern and fossil bone. To detail the conventional procedure for identifying Lines of Arrested Growth (LAGs), we have added Supplementary Information 7 that presents the osteohistological principles followed during our osteohistological analysis as well as through an explanatory diagram of the observed bone growth in our specimens. Although we corroborated the inferred annual cyclicity with a novel suite of techniques (i.e. virtual thick sectioning, stable isotope analysis, and quantified osteocyte lacunar density and volume), the motivation behind these techniques is well understood and completely motivated through published literature.

Line-by-line notes:

Line 27 – you use the word "impact" throughout the abstract/manuscript to mean different things (e.g. "impact on the Yucatan Peninsula" vs. "impacted 76% of species") – a different word/phrase here, e.g. "extirpated" or "vanquished" or more simply "resulted in the loss of" would be both more precise and help reduce repetition and confusion.

✓ corrected

Line 175 – "than e.g. birds" is a bit confusing. Perhaps rephrase "than other groups (e.g. birds)"

✓ corrected

Line 187 – you use birds here again as the only example of surviving lineages. Could expand this a bit to include another group or two (e.g. mammals, fish, etc)

✓ corrected

Reviewer Reports on the Third Revision:

Referee #3 (Remarks to the Author):

During et al present unique evidence using an exceptionally preserved fossil assemblage, to investigate the question of what season the Chicxulub Bolide collided with the earth. I maintain that the approach is novel and the evidence they present, both osteological and geochemical is broadly supportive of their conclusions. Overall I think that this manuscript is ready for publication.